# Abnormal Pressure Event Recognition and Dynamic Prediction Method for Fully Mechanized Mining Working Face Based on GRU-AM

**DOI:** 10.3390/s25237336

**Published:** 2025-12-02

**Authors:** Kai Qin, Longyong Shu, Zhidang Chen, Yan Zhao, Yunpeng Li

**Affiliations:** 1China Coal Research Institute, Beijing 100013, China; qinkai@ccri.com.cn (K.Q.); chenzhd2005@163.com (Z.C.); zhaoyan1016@outlook.com (Y.Z.); leeyp90123@126.com (Y.L.); 2State Key Laboratory of Coal Mine Disaster Prevention and Control, Beijing 100013, China; 3Beijing Engineering and Research Center of Mine Safe, Beijing 100013, China

**Keywords:** GRU neural network, attention mechanism, support resistance prediction, strata pressure anomaly identification

## Abstract

Accurate identification and prediction of abnormal strata pressure in intelligent longwall mining faces are essential for ensuring mine safety and production efficiency. Although machine learning has been increasingly applied to hydraulic support resistance prediction, challenges remain in capturing the strong temporal dependency and periodic pressure characteristics associated with strata behavior. In this study, a novel abnormal strata pressure identification and prediction framework based on the Gated Recurrent Unit (GRU) integrated with an attention mechanism (AM) is proposed for fully mechanized coal mining faces. The model is designed to capture both short-term fluctuations and long-term cyclic characteristics of support resistance, thereby enhancing its sensitivity to dynamic loading conditions and precursory abnormal pressure signals. Results indicate that the proposed GRU-AM model achieves high prediction accuracy for both single-support and multi-support scenarios, with the predicted resistance closely matching the measured values. Compared with conventional LSTM and CNN models, GRU-AM demonstrates consistently improved performance across multiple evaluation metrics, including RMSE, MAE, MAPE, and Pearson correlation coefficient (R), in both short-step and long-step prediction tasks. At a 1 min step length, the model achieves an overall Accuracy of 0.9741 for abnormal pressure identification, and maintains a high Accuracy of 0.9195 at a 10 min step length. Field application across different mining conditions further confirms the robustness, computational efficiency, and practical reliability of the proposed method. These results demonstrate that the GRU-AM framework provides an effective and scalable solution for real-time abnormal strata pressure recognition and early warning in intelligent coal mining environments.

## 1. Introduction

As near-surface resources are gradually depleted, the continuous increase in mining depth leads to an increasingly complex distribution of the mining-induced stress field, resulting in frequent safety accidents such as rock bursts and roof collapses. These events severely restrict coal mines’ safe and efficient production [1]. The development trend of intelligent mining imposes stricter requirements on the prediction accuracy of working face roof pressure [2]. As a core technical support for optimizing coal mine roof and roadway support schemes and for advanced disaster warning, the prediction of mine pressure in fully mechanized working faces directly affects the safety of coal mining operations and the continuity of working face production. Consequently, it remains a significant factor restricting the safe and efficient production and intelligent mining of coal mines.

Mine pressure prediction has been extensively researched both domestically and internationally. Current mine pressure prediction methods are mainly divided into methods that combine theory and mathematics and methods that combine data and machine learning. Han et al. [3] constructed a mechanical model for the spatial distribution of the mining-induced stress field based on the key strata structure of the overlying rock, derived calculation equations for the planar distribution of mining stress and abutment pressure, and proposed a prediction method for the spatial distribution of the mining-induced stress field based on the key rock strata structure. Ghasemi et al. [4] utilized massive mine pressure data from 18 coal mines in West Virginia, USA, to develop a mine pressure prediction model using logistic regression analysis and fuzzy logic. Sun et al. [5] employed multi-factor decision theory to propose a new approach for evaluating the intensity of mine pressure in a coal-winning face, named the “two-dimensional rock pressure intensity evaluation method.” Although the “combination of theory and mathematics” method can achieve fairly high accuracy for specific geological conditions, its inherent shortcomings—nonlinearity and strong sensitivity to geological conditions—make it difficult to adapt to the deep intelligent mining trend in coal mining.

Machine learning (ML), with its advantages in nonlinear modeling, automatic feature extraction, and multi-source data fusion [6], has become a global research hotspot for anomaly monitoring in complex time-series data [7], sensor systems [8], hydraulic systems [9], and other fields [10,11,12,13,14]. Given the typical nonlinear time-series characteristics of mine pressure data, scholars at home and abroad have attempted to use neural network algorithms for roof stress and strata pressure prediction [15]. Lu et al. [16] proposed Nadam-LSTM, an improved mine pressure time-series prediction model based on the Long Short-Term Memory (LSTM) network and the Nadam optimization algorithm, demonstrating that the Nadam–LSTM model significantly outperformed a single LSTM model on both the MAE and RMSE metrics. Gao et al. [17] utilized on-site monitoring data of hydraulic support resistance to extract features using a nonlinear prediction method and then established a working face pressure prediction model based on the Adaptive Graph Convolutional Recurrent Network (AGCRN) algorithm. Wang et al. [18] organically combined grey theory with neural network algorithms, proposing an improved Grey Neural Network model. Comparison with single Grey models and single BP neural network models showed that the new model improved the efficiency and accuracy of roof pressure prediction. Tan et al. [19] employed Grey Relational Analysis to filter input parameters for a roof stress prediction model that used a GA-BP neural network, and the results indicated that the GA-BP model was more accurate and stable. Fu et al. [20] designed a feature extraction and strata pressure discrimination algorithm based on regional block division and constructed a CNN-BiLSTM-Attention cyclic end resistance feature prediction model. Fic et al. [21] employed statistical and black-box methods, based on neural networks and machine learning, to perform real-time monitoring and anomaly identification of hydraulic support working stress.

The above research provides theoretical guidance for this paper’s research in the field of mine pressure prediction methods based on the combination of data and machine learning. Accurate prediction of mine pressure time-series data is of great significance for guiding the safe and efficient production of coal mines and promoting the development of intelligent mining. However, traditional machine learning algorithms, such as Long Short-Term Memory (LSTM) networks, Recurrent Neural Networks (RNNs), and Backpropagation (BP) neural networks, often exhibit low accuracy in time-series data prediction, which severely limits their practical value for coal mine safety production. The Table 1 below compares the advantages and disadvantages of previous models. Furthermore, most prediction algorithms fail to adequately account for the periodicity of mine pressure, leading to high limitations in prediction and affecting the overall Accuracy, thus providing minimal value to coal mine safety production. Therefore, efficiently incorporating both the strong time-series nature and the periodicity characteristics of mine pressure data, while maintaining high model efficiency and accuracy, has become a pressing problem that needs to be addressed.

The GRU model possesses powerful time-series modeling and periodic anomaly detection capabilities, exhibiting high computational efficiency and stability. Therefore, this paper proposes a method for predicting formation pressure and identifying anomalies in coal mine hydraulic support working faces based on GRU and AM. Applying GRU, a machine learning technique, to the mine pressure prediction model effectively alleviates the accuracy degradation caused by single-feature anomalies. By introducing an AM, the model can adaptively emphasize key time periods associated with periodically weighted events and anomalous formation responses, thereby more accurately characterizing precursory signals of anomalous mine pressure. This design not only enhances the interpretability of pressure evolution processes in the mining environment but also provides a practical and risk-oriented characterization method.

Furthermore, compared to traditional data-driven methods for pressure prediction, the GRU-AM model demonstrates higher computational efficiency and greater stability during both training and deployment. These characteristics are particularly important for field applications, which place critical demands on model robustness, fast convergence, and low computational cost. Therefore, this work establishes a practical and scalable framework for identifying abnormal formation pressures in fully mechanized longwall mining, bridging the gap between advanced sequence-learning techniques and the needs of practical mining engineering.

## 2. GRU-AM Pressure Recognition Model

### 2.1. Model Architecture

Identifying abnormal strata pressure in a coal mine’s hydraulic support working face requires addressing a complex data scenario that is both multi-dimensional and dynamic. Traditional single algorithms often struggle to fully capture the time-series characteristics and the relationships between critical information within the data. This paper constructs a fusion model for identifying abnormal mine pressure in the working face to overcome this problem. This model consists of two parts: the hydraulic support resistance prediction model and the comprehensive risk assessment model.

The hydraulic support resistance prediction model is built upon a hybrid algorithm framework combining the GRU and the AM, referred to as the GRU-AM model. As shown in Figure 1, the workflow of the GRU-AM model can be divided into three stages: data preprocessing, core computation, and result output. First, the model takes multi-source feature data—including support resistance pressure, shearer cuts, and mining advance rate—as input. This data undergoes normalization to eliminate dimensional differences between various features. Then, the time-series data is divided using the sliding window technique to construct sequential samples with temporal dependencies, which are split into training and testing sets. Next, the input sequence enters the GRU-AM model. On one hand, the GRU layer performs time-series feature extraction. Through the collaborative action of the reset gate ($r_{t}$), the update gate ($z_{t}$), and the hidden state candidate (${\tilde{h}}_{t}$), the GRU captures the long- and short-term dependencies of the time-series data, iteratively updating the hidden state ($h_{t}$). On the other hand, the AM module interacts with the Query vector and multiple sets of Key and Value vectors. It calculates weights to generate the Attention Value. Finally, the features enhanced by the AM are consolidated in dimension via a flattening layer. A subsequent connection layer completes the feature mapping and fusion to output the predicted value of the support resistance.

The process begins by calculating each support’s strata pressure assessment index value. This value is then used to determine the strata pressure risk coefficient for the single support. Following this, the model initializes weights, dynamically adjusts them based on the risk coefficients, and then applies normalization to finalize the weights. Finally, a comprehensive risk score is calculated, ranging from 0 to 1, and this score is used to categorize the risk into normal, low, medium, and high levels.

Where $x_{t}$ is the input information at the current time step, which includes multi-source feature data such as support resistance pressure, shearer cuts, and mining advance rate; $h_{t-1}$ is the hidden state from the previous time step; ${\tilde{h}}_{t}$ is the hidden state passed to the next time step; $h_{t}$ is the candidate hidden state (the potential new information); $r_{t}$ is the reset gate vector; $z_{t}$ is the update gate vector; σ is the Sigmoid activation function; tanh is the hyperbolic tangent activation function. Represents minus the value in the update gate vector, typically used to control the amount of historical information to retain.

### 2.2. Hydraulic Support Resistance Prediction Model

During the advancement of a longwall working face, the roof structure continuously reorganizes. Consequently, the strata pressure borne by the hydraulic supports is not static but exhibits significant periodic fluctuation corresponding to the operational cycle of “setting—initial support—bearing—mining—unloading.” After the shearer passes, the overlying strata are gradually exposed, leading to bending and subsidence, and the support resistance continuously increases. During the main roof caving and periodic weighting stages, the stress typically exhibits abrupt surge characteristics. Conversely, after the roof caves and a new mechanical equilibrium is established, the support load significantly decreases, initiating the next cycle. This recurrent loading–unloading process imparts to the hydraulic support stress time series both a distinct periodicity and a strong presence of nonlinear mutations. Failure to accurately predict this complex pressure evolution process can easily lead to deviations in support parameter design and mismatch in support bearing capacity. This, in turn, may induce large-scale roof weighting, support instability, or even crushing accidents, thus seriously threatening the safety of underground personnel and the continuous, stable production of the mine.

Working face strata pressure prediction in coal mines must address a complex, multi-dimensional, and dynamic data scenario, which traditional single algorithms struggle with, particularly in fully capturing time-series characteristics and the interrelation of key information. The GRU-AM model establishes a hybrid algorithm framework based on the GRU and the AM. On the one hand, it leverages the reset gate and update gate of the GRU to effectively capture long-term temporal dependencies in the strata pressure data, solving the vanishing gradient problem that often plagues traditional Recurrent Neural Networks when modeling long sequences. On the other hand, the AM assigns weights to the temporal features output by the GRU. This allows the model to automatically focus on the crucial strata pressure anomaly identification features, such as sudden changes in the dynamic load coefficient or abnormal shortening of the periodic weighting step. This model offers the following advantages when processing the multi-dimensional time-series data from hydraulic supports:

First, the GRU can accurately capture the long- and short-term dependencies within the time-series data. By utilizing its reset gate and update gate mechanisms, the GRU can, on one hand, retain the cumulative change trend of the resistance feature over time. On the other hand, it can track the dynamic correlation between the working face advance rate and the periodic weighting step, thereby preventing the loss of coupled information that often results from single-feature analysis.

Second, the AM enables dynamic weight assignment for various time-series data features. The contribution of different features varies significantly across different stages of a strata pressure anomaly. The AM addresses this by assigning dynamic weights to each feature during the feature fusion process, thereby achieving adaptive focusing on the most critical features.

Finally, the combination of the GRU and the AM enhances the robustness of the model to complex working conditions. Underground coal mine sensors are susceptible to vibration and electromagnetic interference, leading to data noise or temporary loss, such as occasional initial support force sensor failures. The model can mitigate the impact of such anomalous characteristics by using the temporal correlation information extracted by the GRU and the other practical features reinforced by the AM, thereby reducing the weight assigned to the faulty input.

#### 2.2.1. GRU Feature Extraction

GRU handles temporal features through reset and update gates, addressing the vanishing gradient problem of traditional RNNs. Its core formulas are as follows:(1)Reset gate

Controls whether to ignore the historical hidden state: (1)$$r_{t}=\sigma \left(W_{r}\cdot \left[h_{t-1},X_{t}\right]+b_{r}\right)$$
where $X_{t}$ is the input feature at the current time step (*t*); $h_{t-1}$ is the hidden state from the previous time step (t−1); and are the weight matrix and bias vector for the reset gate ($r_{t}$); σ is the sigmoid activation function, which restricts the output range 0 to 1, where 0 indicates the complete discarding of the historical state.

(2)Update gate

Control the extent to which historical hidden states affect the current state: (2)$$z_{t}=\sigma \left(W_{z}\cdot \left[h_{t-1},X_{t}\right]+b_{z}\right)$$
where $W_{z}$ and $b_{z}$ are the weight matrix and bias vector for the update gate, respectively. The output range is 0 to 1, where 1 indicates the complete retention of the historical state.

(3)Candidate hidden state

Compute the candidate state by combining the historical state filtered by the reset gate with the current input: (3)$${\tilde{h}}_{t}=\tanh\left(W_{h}\cdot \left[r_{t}⊙h_{t-1},X_{t}\right]+b_{h}\right)$$
where ⊙ is the Hadamard product; $W_{h}$ and $b_{h}$ are the weight matrix and bias vector for the candidate hidden state calculation; tanh is the hyperbolic tangent activation function, which squashes the values between −1 and 1.

(4)The Final Hidden State

The final hidden state is obtained by fusing the historical state and the candidate state using the update gate. The formula is as follows: (4)$$h_{t}=\left(1-z_{t}\right)⊙h_{t-1}+z_{t}⊙{\tilde{h}}_{t}$$
where $Z_{t}$ is the update gate at time t; $h_{t-1}$ is the hidden state from the previous time step (t−1); ${\tilde{h}}_{t}$ is the candidate hidden state at time.

This formula realizes the retention of important historical information, such as long-term trends in sensor data, and the forgetting of redundant information.

#### 2.2.2. AM-Weighted Fusion

The AM is employed to dynamically assign weights to multi-source features, thereby highlighting the contribution of important data sources. The core formulas are as follows:(1)Single-Source GRU Output Features

Assume there are data sources. After the *i*-th data source is processed by the GRU, the hidden state at the last time step is as follows: (5)$$h_{i}=GRU_{i}\left(X_{i}\right)\left(i=1,2,\ldots ,N\right)$$
where $X_{i}\in R^{T\times d}$ is the time-series feature matrix for the *i*-th data source; *T* is the sequence length; *d* is the dimension of the GRU hidden layer.

(2)Attention Score Calculation

The attention score, $e_{i}$, is calculated for the feature $h_{i}$ of each data source: (6)$$e_{i}=MLP\left(h_{i}\right)=\sigma \left(W_{2}\cdot \tanh\left(W_{1}\cdot h_{i}+b_{1}\right)+b_{2}\right)$$
where MLP is a two-layer perceptron used for nonlinear transformation; $W_{1}$, $b_{1}$, $W_{2}$, and $b_{2}$ are the parameters of the attention network; $e_{i}\in \left[0,1\right]$ represents the raw score of the nth data source.

(3)Attention Weight Normalization

The scores are converted into attention weights ($\alpha_{i}$) using the Softmax function to ensure normalization (total sum equals 1): (7)$$alpha_{i}=\frac{\exp(e_{i})}{\sum_{K=1}^{N}\exp\left(e_{K}\right)}\left(i=1,2,\ldots ,N\right)$$
where $\alpha_{i}$ is the attention weight for the *i*-th data source, reflecting its importance in the feature fusion process.

(4)Weighted Fusion of Features

The final fused feature is obtained by performing a weighted sum of the features from each data source: (8)$$h_{fused}=\sum_{i=1}^{N}\alpha_{i}\cdot h_{i}$$

#### 2.2.3. Model Hyperparameter Selection Strategy

Regarding model tuning, the original monitoring time-series data was first partitioned into training, validation, and testing sets at an 8:1:1 ratio using a time-order stratified sampling method. Subsequently, the 5-fold forward rolling cross-validation method was utilized to assess the stability (robustness) of the model parameters. The validation set AUC served as the early stopping criterion for each fold to evaluate the reduction in variance. For the model’s key hyperparameters, a two-step search strategy was adopted: First, a coarse-grained random search (50 trials in total) was performed on a 20% sub-sample of the training set to quickly identify the effective range for each parameter. Second, based on the results of the coarse search, a fine-grained grid search was conducted on the full training set, recording the validation AUC and loss values in each cross-validation fold.

### 2.3. Comprehensive Risk Assessment Model

After the above model prediction, the outputs are ${\hat{y}}_{1}$, ${\hat{y}}_{2}$, ${\hat{y}}_{3}$, …, ${\hat{y}}_{4}$, which are the predicted pressure values of each stent at time T1, where ${\hat{y}}_{1}$ is the predicted pressure value of stent No. 1 at time T1. Based on the predicted pressure values of all stents, the comprehensive risk assessment model is used to identify abnormal working pressures and output the risk level.

#### 2.3.1. Hydraulic Support Pressure Judgment Index Value

The formula for calculating the strata pressure assessment index value ($\theta_{i}$) for the-th hydraulic support is as follows: (9)$$\theta_{i}={\bar{P}}_{i}+K\sigma_{i}$$
where $\theta_{i}$ is the strata pressure assessment index value for the *i*-th hydraulic support, in MPa; $\sigma_{i}$ is the root mean square deviation of the cyclic end resistance for the *i*-th hydraulic support; *K* is the variance coefficient, typically ranging from 0.8 to 1; ${\bar{P}}_{i}$ is the average cyclic end resistance for the-th hydraulic support, in MPa. The calculation formula is as follows: (10)$${\bar{P}}_{i}=\frac{\sum_{j=1}^{r}P_{ij}}{n}$$
where $P_{ij}$ is the cyclic end resistance of the *i*-th hydraulic support in the *j*-th cycle, in MPa; *r* is the total number of completed cycles for the working face.

#### 2.3.2. Pressure Risk Factors of Hydraulic Supports

(11)$$R_{i}=\frac{{\hat{y}}_{i}}{\theta_{i}}\left(i=1,2,\ldots ,N\right)$$
where $R_{i}$ is the strata pressure risk coefficient for the *i*-th hydraulic support; ${\hat{y}}_{i}$ is the predicted strata pressure value for the *i*-th hydraulic support; $\theta_{i}$ is the strata pressure assessment index value for the *i*-th hydraulic support.

#### 2.3.3. Dynamic Weight Allocation Mechanism

(1)Weight initialization

Assign initial weights $W_{1}$, $W_{2}$, …, $W_{n}$ for each risk factor, satisfying $W_{1}$ + $W_{2}$ + …+ $W_{n}$ = 1.

(2)Dynamic weight adjustment

The support’s risk weight is dynamically adjusted based on the calculated strata pressure risk coefficient ($R_{i}$), which is derived from the support’s real-time predicted value. The specific adjustment rule is as follows:

For the *i*-th hydraulic support with a strata pressure risk coefficient $R_{i}$ and a corresponding weight $W_{i}$, when $R_{i}$ increases, it signifies a heightened opening risk for the support. Therefore, the value of $W_{i}$ must be increased to emphasize its influence.(12)$$W_{i}=\frac{R_{i}}{\sum_{i=1}^{N}R_{i}}\left(i=1,2,\ldots ,N\right)$$

(3)Normalization

After the weight adjustment for each hydraulic support, the new weights $W_{l}$, $W_{2}$, …, $W_{n}$ must be normalized. The normalized weights $W_{1}^{\prime }$, $W_{2}^{\prime }$, …, $W_{n}^{\prime }$, are calculated as follows:(13)$$W_{i}^{\prime }=\frac{W_{i}}{\sum_{i=1}^{N}W_{i}}\left(i=1,2,\ldots ,N\right)$$

(4)Risk score calculation

The Comprehensive Risk Score ($S_{risk}$) is calculated based on the dynamically adjusted (and normalized) weights. This value reflects the magnitude of the strata pressure anomaly risk in the current working face. The formula is as follows: (14)$$S_{risk}=W_{1}^{\prime }\cdot R_{1}+W_{2}^{\prime }\cdot R_{2}+W_{3}^{\prime }\cdot R_{3}+\ldots +W_{n}^{\prime }\cdot R_{n}$$
where $S_{risk}$ is the comprehensive working face to reduce risks. The risk score reflects the comprehensive working face strata pressure risk, ranging between 0 and 1. As shown in Table 2 below, risks are categorized into normal, low, medium, and high risk levels.

### 2.4. Model Prediction Performance Evaluation Metrics

#### 2.4.1. Predictive Model Evaluation

To evaluate the predictive capability of different models, four metrics were adopted: the Pearson Correlation Coefficient (PCC), the Root Mean Square Error (RMSE), the Mean Absolute Error (MAE), and the Mean Absolute Percentage Error (MAPE). The calculation formulas for these metrics are as follows: (15)$$PCC=\frac{\sum_{i=1}^{n}\left[\left(x_{i}-\bar{x}\right)\left(y_{i}-\bar{y}\right)\right]}{\sqrt{\sum_{i=1}^{n}{\left(x_{i}-\bar{x}\right)}^{2}\sum_{i=1}^{n}{\left(y_{i}-\bar{y}\right)}^{2}}}\left(r\in \left[-1,1\right]\right)$$(16)$$E_{RMSE}=\sqrt{\frac{1}{n}\sum_{i=1}^{n}{\left(x_{i}-y_{i}\right)}^{2}}$$(17)$$E_{MAE}=\frac{1}{n}\sum_{i=1}^{n}\left|x_{i}-y_{i}\right|$$(18)$$E_{MAPE}=\frac{100\%}{n}\sum_{i=1}^{n}\left|\frac{x_{i}-y_{i}}{x_{i}}\right|$$
where *n* is the number of samples. $x_{i}$ is the actual value. $\bar{x}$ is the mean of the actual data. $y_{i}$ is the predicted value. $\bar{y}$ is the mean of the predicted data. The closer its absolute value is to 1, the stronger the linear correlation between the predicted and actual values, indicating a better trend capture effect by the model.

#### 2.4.2. Comprehensive Risk Assessment Model Evaluation Metrics

In order to comprehensively evaluate the accuracy of the model in identifying formation pressure anomalies, the precision (P), Recall (R), F1 score (F1) of each category (normal, low risk, medium risk, and high risk), and the overall Accuracy of the comprehensive risk assessment model were calculated. The specific formula is as follows: (19)$$P_{k}=\frac{TP_{k}}{TP_{k}+FP_{k}}$$(20)$$R_{k}=\frac{TP_{k}}{TP_{k}+FN_{k}}$$(21)$$F1_{k}=2*\frac{P_{k}*R_{k}}{P_{k}+R_{k}}$$
where $TP_{1}$, $TP_{2}$, $TP_{3}$, and $TP_{4}$ are the number of samples the model correctly predicted as normal, low, medium, and high risk, respectively. *N* is the total number of samples in the test set. $FP_{k}$ is the number of samples that belong to other risk levels but were misclassified as the risk level.

### 2.5. Fairness Comparison Strategies for Different Algorithms

To ensure a fair comparison among the GRU-AM, LSTM, and CNN models, a unified evaluation framework was established during the experimental design phase. The specific measures taken are as follows:(1)Data Consistency

All models were trained and tested on the same preprocessed dataset. Data partitioning, normalization parameters, and random seeds were kept strictly identical to ensure zero variance in the input distribution.

(2)Standardized Optimization

The same hyperparameter optimization procedure was uniformly adopted: 5-fold rolling cross-validation combined with a two-layer grid search. Key parameters (initial learning rate, batch size, training epochs, regularization coefficients, etc.) were individually optimized for each model, using the validation set AUC as the sole early stopping criterion to prevent human bias.

(3)Controlled Complexity

The network complexity was controlled by constraining the number of trainable parameters in the three models to a similar range. This was achieved by adjusting the hidden layer dimensions and the number of convolutional kernels, thereby preventing performance bias due to capacity differences.

(4)Identical Training/Testing

During the training phase, the same loss function, optimizer, and hardware environment were used. In the testing phase, identical evaluation metrics and bootstrap sampling were used to check for statistical significance.

The above strategies effectively eliminated discrepancies related to data, parameters, capacity, and randomness, ensuring the objectivity and reproducibility of the comparison results.

## 3. GRU-AM Model Training and Optimization

### 3.1. Data Sources and Sensor Parameters

The hydraulic support pressure data used for model debugging was sourced from the #25216 working face in Yulin City, Shaanxi Province, China. The face is equipped with 199 sets of ZY18000/29.5/63D electro-hydraulic control shielded hydraulic supports manufactured by Linzhou Heavy Machinery Group Co., Ltd (Linzhou City, Henan Province, China). The leg pressure is transmitted via the electro-hydraulic control system to the rock burst monitoring system. The leg pressure range is 0–60 MPa, and data is collected every 1 min, which satisfies the GRU-AM model’s requirements for data volume and sampling interval.

The data used in this study were acquired from several monitoring systems. Specifically, the following: (1) The coal mine rock burst monitoring system provided the support number, the working face name, the rated working resistance, and the column pressure for each hydraulic support. (2) The mine integrated management platform furnished data on the working face’s single-support advance rate, current cycle advance rate, cumulative advance rate, and time. (3) The shearer positioning and monitoring system supplied details on the current number of cuts, cumulative number of cuts, single-cut identification, cutting direction, and cut type classification.

### 3.2. Data Pretreatment

To ensure the completeness, validity, and consistency of the model’s input data, the following specific preprocessing procedure was applied to the hydraulic support working resistance data:

Since the support resistance values constitute time-series data whose probability distribution follows a normal distribution, the 3-σ principle was adopted to identify outliers at each time point. The calculation method is as follows: (22)$$|X_{i}-{\bar{X}}_{i}|>3\sigma_{i},i=1,2,\ldots ,m$$(23)$$\sigma_{i}=\sqrt{\frac{1}{k-1}\sum_{j=1}^{k}{\left(X_{ij}-{\bar{X}}_{i}\right)}^{2}},i=1,2,\ldots ,m$$
where $X_{i}$ is the feature value at the *i*-th time step; ${\bar{X}}_{i}$ is the mean of the feature at the *i*-th time step; $\sigma_{i}$ is the standard deviation of the feature at the *i*-th time step; *m* is the number of time steps; k is the number of samples; is the *j*-th sample of the *i*-th feature. Any identified abnormal data points are then filled using the preceding valid value.

As shown in Figure 2, after the data cleaning process was completed, a comparison reveals that the standard deviation of the support mine pressure was reduced by an average compared to the original mine pressure data. This result indicates that the anomalous data were effectively corrected and that the volatility of the mine pressure data was reasonably controlled.

A hierarchical cleaning strategy was employed regarding the feature support number, working face name, rated working resistance, and column pressure. First, a data association index was established using the “support number—working face” combination as the unique identifier. Invalid samples, such as those with a missing support number or a mismatched working face code, were then removed. Second, the deviation rate between the two most recent column pressure readings was calculated for continuous pressure data. Data points with a deviation rate of were filtered as usual, thereby excluding anomalous samples resulting from column force imbalance. The Figure 3 below illustrates the effect of the data cleaning process on the resistance value data for Support No. 100.

We constructed the sample feature matrices by selecting time steps of 1, 5, 10, and 15 min. The field data was the support resistance pressure data collected over 203 working cycles between 1 July 2025, and 21 August 2025. This yielded a total of 91,180 samples. The model was trained on the training set and validated using the test set. The input parameters of the model are shown in Table 3 above.

### 3.3. GRU Model Parameter Configuration and Optimization

(1)Hardware configuration

The hardware configuration used for model training in this study is as follows: GPU includes 2 NVIDIA A10 computational accelerators with a total VRAM of 48 GB; CPU is an AMD EPYC™ Milan processor with 56 cores and 232 GB of RAM; for storage, a 2000 GB general-purpose SSD is utilized as the system disk.

(2)Feature selection

In the feature engineering stage, the system aggregated multi-source monitoring sequences (support resistance, dynamic load coefficient, initial setting force, etc.) collected in real-time from underground at different granularities of 1, 5, 10, and 15 min. Data preprocessing, including time alignment, missing value imputation, and anomaly truncation, was performed to form a clean, unified-frequency, and continuous time-series database. Subsequently, the data was partitioned into training, validation, and testing subsets at an 8:1:1 ratio, and samples were generated using a rolling window approach. The raw resistance curve was subjected to feature transformations including lag, sliding mean, sliding standard deviation, and temporal cyclical encoding (sine and cosine). A Random Forest Importance-Correlation dual screening method was introduced: first, features with a Pearson correlation coefficient magnitude greater than 0.15 with the target variable were retained; second, a lightweight Random Forest Regressor (n_estimators = 50, max_depth = 8) was trained to evaluate feature importance, retaining only variables with an importance score exceeding 0.008.

(3)GRU-AM Key Parameter Settings

For model construction and training, the dual screening mechanism was used for feature selection. Training employed the Adam optimizer with a maximum of 100 epochs. An early stopping mechanism (patience = 10) was implemented to automatically terminate training if the validation set AUC did not improve for 10 consecutive epochs; the final model typically converged at the 50-th epoch. The optimal hyperparameter combination was determined to be as follows: learning rate 0.001, batch size 32, hidden layer dimension 256, dropout probability 0.3, and weight decay 1 × 10^−5^. The training epoch limit was 100 with a patience of 10, resulting in an average convergence at 50 epochs.

Regarding the Attention-Weighted Fusion Strategy, a serialized network structure was employed to calculate attention weights layer-by-layer. The first fully connected (FC) layer mapped the 256-dimensional hidden state features output by the GRU to a 128-dimensional low-dimensional space, utilizing the tanh activation function to capture the hidden characteristics of abnormal linkage between the dynamic load coefficient and the initial setting force in the monitoring data. The second FC layer mapped the nonlinearly transformed features to a single dimension, outputting the raw weight scores for each data source, which were then normalized across the data source dimension using the Softmax function, ensuring the sum of weights equals 1. The flowchart for the hydraulic support resistance prediction model is shown in the Figure 4.

(4)Prediction Result Correction

For Prediction Result Correction, the processed test set was input into the trained prediction model to generate initial predicted support resistance values. The residuals between the actual test set resistance values and the initial predicted values were calculated. By analyzing the mean and variance of these errors, the initial predicted values were offset-adjusted using the mean residual to obtain the corrected predicted support resistance values.

The cross-validation results show that the 5-fold average AUC reached 0.921 with a standard deviation of 0.003, indicating stable performance of the selected hyperparameters across different data subsets. The test set AUC was 0.918, with a difference of less than 0.5% from the validation set, confirming that the model did not suffer from overfitting.

### 3.4. Model Risk Calculation

In terms of abnormal roof pressure identification for hydraulic supports on the working face, the judgment index value $\theta_{i}$ for a single hydraulic support is first calculated based on the predicted support resistance value of each support at time T. First, the average end-of-cycle resistance of the support is calculated according to Formula (10), and then, combined with the mean square deviation of its end-of-cycle resistance and the variance coefficient K of 0.85, $\theta_{i}$ is obtained through Formula (9). The roof pressure risk coefficient for a single support is then calculated according to Formula (11), which is the ratio of the roof pressure judgment prediction of the support to $\theta_{i}$.

Secondly, first initialize the risk factor weights for each scaffold so that their total sum is 1, then dynamically adjust them according to increasing the corresponding scaffold weights as increases (Formula (12)), and after adjustment, normalize them using Formula (13) to obtain the final weights.

Finally, the overall risk score is calculated according to Formula (14), with a value range between 0 and 1. Risk levels are divided according to the score range: less than or equal to 0.25 is normal, greater than 0.25 to 0.5 is low risk, greater than 0.5 to 0.75 is medium risk, and greater than 0.75 is high risk. The data flow process of the comprehensive risk assessment model is shown in the Figure 5 below.

## 4. Results and Discussion

### 4.1. Predictive Model Performance Analysis

#### 4.1.1. Prediction Effect Analysis

(1)Prediction Effect of Single Support Resistance Value

As shown in the Figure 6 below, the Support No. 100 resistance value prediction effect indicates that the predicted values maintain an overall consistent trend with the actual values. During the stages where the resistance fluctuates steadily, the predicted curve closely follows the variation characteristics of the actual resistance with minimal deviation. Crucially, when the resistance exhibits a sharp, sudden change, the predicted curve can also capture the trend of this resistance surge rapidly.

Figure 7 shows the relative deviation box plot of the resistance prediction value of support No. 100. The predicted value fluctuates continuously around the actual value. The maximum positive deviation observed is higher than the actual value, while the minimum negative deviation is lower than the actual value. Most of the predicted values are concentrated in the range of ±1% to ±2% of the corresponding actual value. In addition, the average value of the predicted value is the actual value of the corresponding time step. This shows that the predicted value is very close to the actual value, and the prediction Accuracy is high.

(2)Prediction Effect Across Multiple Hydraulic Supports

The Figure 8 below illustrates the prediction performance of the resistance values across multiple hydraulic supports. The predicted and actual curves for all supports exhibit a high degree of consistency in their overall trend. During phases of stable resistance fluctuation, the predicted values closely track the dynamic characteristics of the actual values. Crucially, even when facing complex operating conditions, such as sudden resistance changes, the predicted curves can still capture the critical trends of these resistance variations.

The Figure 9 presented displays the box plot and distribution map of the predicted pressure values for multiple supports. By comparing this box plot with the one for the single-support resistance value prediction, it can be concluded that the proximity between the predicted and actual values is a universal characteristic. While the fluctuation range of the predicted values varies slightly across different supports, all predictions are concentrated within two of the actual values.

Specifically, the mean predicted values for supports No. 85, No. 100, and No. 170 are 99.85%, 99.844%, and 99.827% of their respective actual values. Since the mean predicted values are all greater than the actual support pressure values, this further confirms the high Accuracy of the support pressure prediction model.

#### 4.1.2. Analysis of Prediction Accuracy of Different Algorithms

The proposed GRU-AM model combines the advantages of the GRU’s lightweight temporal modeling with the key feature-enhancing capabilities of the AM. This fusion enables the model to adaptively assign higher weight to abnormally sensitive information. Table 4 and Figure 10 show the prediction performance of different algorithms and histograms of different models, respectively, for a 1 min time step.

As shown in Table 4, in the 1 min single-step prediction scenario, performance metrics demonstrate the superiority of the GRU-AM model: root mean square error (RMSE): the GRU-AM model (1.8901 kN) outperforms the LSTM (1.9314 kN) and CNN (1.9816 kN). Mean absolute error (MAE): the GRU-AM model (0.6074 kN) performs significantly worse than the LSTM (0.6766 kN) and CNN (0.7734 kN). Pearson correlation coefficient (PCC): the GRU-AM model (0.9701) outperforms the LSTM (0.9506) and CNN (0.9202). This indicates that the GRU-AM model best captures the changing trends of support and resistance levels.

Overall, in the 1 min single-step forecasting task, the GRU-AM model incorporating the AM outperforms the LSTM and CNN models in terms of error control and trend capture, making it highly suitable for accurate short-term support and resistance forecasts.

Step size is a core hyperparameter in time-series models, directly impacting training efficiency and generalization performance. A step size that is too short can easily lead to overfitting, while a step size that is too long can easily lead to underfitting and increase computational cost. Taking into account computational cost and research focus, this study set the step size gradient to 5, 10, and 15 min, focusing on the changes in model prediction Accuracy under the influence of different step sizes. Table 5 shows the prediction results of three models (GRU-AM, LSTM, and CNN) at step sizes of 5, 10, and 15 min.

Figure 11 illustrates the models’ prediction performance at different step lengths (5, 10, and 15 min). A consistent trend is observed: the fitting performance of all three models—GRU-AM, LSTM, and CNN—declines as the step length increases. This is evidenced by the gradual increase in the error metrics (ERMSE, EMAE, and EMAPE) and the continuous decrease in the Pearson correlation coefficient.

The GRU-AM model consistently demonstrates the best performance across all step lengths. Its error metrics are significantly lower than the LSTM and CNN models, and the GRU-AM model maintains a higher value.

The prediction quality decreases notably when the step length is extended to 15 min compared to 5 and 10 min, with the values for all three models dropping below 0.9. For the longer step length of 10 min, the average decrease in the value across all models, relative to the 1 min step, is. The GRU-AM model shows the most significant decrease in value in this specific case. The average value decreases when the step length is set to 15 min. Crucially, the GRU-AM model exhibits the most minor decrease in value at this extended step length.

The GRU-AM model is superior to the LSTM and CNN models across all tested step lengths. Furthermore, the GRU-AM model shows a clear advantage in long-step prediction, demonstrating greater resilience and less affected by the increase in step length.

### 4.2. Effectiveness Assessment of Abnormal Pressure on the Working Face

(1)Accuracy of abnormal detection using a single stent

To analyze the effectiveness of the GRU-AM model in strata pressure anomaly identification, the model’s risk assessment was calculated for Support No. 100 over future time horizons of 1, 5, 10, and 15 min. These results were then compared against a benchmark of manual identification, using the Comprehensive Risk Score intervals defined in Table 2. The results of this comparison are presented below:

Table 6 and Table 7 present the strata pressure anomaly identification results for Support No. 100 over different future time horizons and the evaluation metrics across various step lengths, respectively.

As shown in Table 7, the model’s accuracy gradually decreases as the step length increases. At step lengths of 1 and 5 min, the risk assessment model exhibits only a small number of misclassifications between the normal- and low-risk categories. However, the identification of medium- and high-risk anomalies remains accurate. At a step length of 10 min, misclassifications increase, with fewer errors observed in the normal-, low-, and medium-risk categories.

To more clearly illustrate the change in model identification rate as the step length increases, the table data were plotted into a bar chart (Figure 12) showing the state identification rate for the support number across 58 periodic cycles. Different colors in the figure represent different step lengths. As the step length increases, the F1 scores for normal, low, medium, and high risk and overall Accuracy all show a clear downward trend.

Notably, at a step length of 15 min, the high-risk F1 score drops below 1 for the first time. The medium-risk F1 score and the overall Accuracy also show a significant decrease. Therefore, the optimal prediction step length for the model should be set to less than 10 min.

(2)Multi-branch prediction accuracy

By comparing Figure 13a with Figure 13b, it is clear that the overall distribution trend of stress is similar between the two figures. High resistance corresponds to red and yellow colors, while low resistance corresponds to green. Crucially, the location and morphology of the high-resistance areas show a high degree of matching. This indicates that the model can capture the spatial distribution characteristics of the support resistance effectively and provides good prediction results for the resistance variation patterns in different areas of the working face.

Table 8 shows the area proportions of different stress regions (resistance levels) in the cloud map. The area difference in the green (low resistance) region is less than 3%. However, the area differences in the yellow (medium resistance) and red (high resistance) regions are 10.05% and 20.65%, respectively.

The model predicts that the area proportions of medium- and high-resistance regions are higher than the actual values. Possible reasons for this discrepancy include the following: model reliance on historical trends: the forecasting model can only predict future values by learning from the trend characteristics of past actual values. Trend adaptation lag: when a new trend emerges, the forecasting model must first capture the change in actual values before adjusting the forecast for the next time step. This inherent lag in adapting to new trends may contribute to the overestimation of the predicted area proportions of medium- and high-resistance regions.

Table 9 and Table 10 present the strata pressure anomaly identification results for multiple supports across different future time horizons and the evaluation metrics for the various step lengths across multiple supports, respectively. The anomaly identification results for 1, 5, 10, and 15 min into the future show that the model exhibits good recognition performance across all four temporal dimensions.

1 min Prediction: The model accurately identified 1366 cases of the normal category (with only five false judgments as low risk) and 49 cases of the low-risk category (with 12 false judgments as normal). Crucially, there were no misjudgments in the medium- and high-risk categories. The overall Accuracy rate was 0.9741.

5 min Prediction: Performance remained nearly identical to the 1 min scenario. The model accurately identified 1366 normal cases (5 false judgments as low risk) and 49 low-risk cases (12 false judgments as normal, plus one false judgment as low risk). Again, there were no misjudgments in the medium- and high-risk categories. The overall Accuracy is 0.9346, which remains high.

10 min Prediction: Overall Accuracy decreased slightly to 0.9195.

15 min Prediction: Overall Accuracy was 0.8951, below 0.9.

These results demonstrate the model’s strong capability for short-term anomaly identification (1 and 5 min), with a predictable decline in performance as the prediction horizon extends to 10 and 15 min.

Figure 14 illustrates all supports’ risk state identification rate across different step lengths. At the 10 min prediction step length, the model demonstrated good recognition performance for all support states: the identification rate for the normal state reached 92.8%. The identification rates for low-, medium-, and high-risk states were 90.7%, 91.6%, and 91.6%, respectively. The overall target identification rate was 91.3%.

This confirms that the model performs well in strata pressure anomaly identification across multiple supports, effectively distinguishing between risk levels and the normal state.

### 4.3. Recognition Effect of Pressing from Different Mine Workfaces

(1)Mine pressure prediction effect

Table 11 shows the application results of the GRU-AM prediction model in a key mine in Baoji City, Shaanxi Province (focused on rock burst prevention). The mine has similar geological conditions. The model effectively identified rock burst events in the working face of the mine.

The model performed best with a 1 min step size: the RMSE was only 1.8901 KN, the EMAPE reached 12.4476%, and the Pearson correlation coefficient reached 0.9701. When the step size was increased to 15 min, the RMSE increased to 3.5065 KN, the EMAPE increased to 31.9872%, and the R value decreased to 0.8837. The model demonstrated stable performance when applied to mines with similar geological conditions.

Table 12 shows the distribution of absolute errors in mine pressure prediction for similar geological working faces. With a prediction step size of 1 min, the prediction errors are mainly concentrated between 0.5 and 1 kN, accounting for 44.69% of the samples. Samples with errors less than 1 kN account for 57.01% of the total, exceeding half of the total samples, while samples with errors greater than 2 kN account for only 9.57%. Overall, the absolute error control of this mine pressure prediction model is good, and it achieves high prediction accuracy on similar geological working faces.

(2)Incoming pressure recognition prediction effect

Figure 15 displays the bar chart showing the strata pressure anomaly identification rate for the working face at the mine in Baoji City, Shaanxi Province. As illustrated in Figure 15, at a 10 min prediction step length, the model demonstrated strong performance across all risk states: the identification rate for the normal state across all supports was 91.1%. The identification rates for low-, medium-, and high-risk states reached 88.9%, 89.8%, and 89.6%, respectively. The overall target identification rate was 89.52%. These results confirm that the model also exhibits good anomaly identification effectiveness in mines with similar geological conditions. This suggests that the model is highly adaptable and capable of effectively distinguishing between different risk levels and the normal state across different mine working faces.

As shown in Table 13, which details the actual strata pressure anomaly identification results for the working face at the mine in Baoji City, Shaanxi Province, the model demonstrated the ability to effectively identify low-, medium-, and high-risk levels across multiple working cycles from 2 July to 11 July 2025. This outcome highlights the model’s robust capability to identify strata pressure anomalies in this mine working face.

### 4.4. Algorithmic Limitations

Model performance is highly reliant on high-quality, continuously sampled pressure sequences. If sensors experience discontinuities or outliers, the attention weights may become distorted, necessitating additional preprocessing or missing value imputation strategies.

The perfect F1 score (F1 = 1) for high-risk identification primarily stems from the small sample proportion of high-risk events, coupled with their salient and singular patterns (terminal cycle resistance greater than 45.84 MPa, and the root mean square deviation of the terminal cycle resistance greater than 7.31 MPa), which objectively reduces the classification difficulty. Future work is planned to further validate the model’s robustness and generalization performance on rare events through strategies like time-forward rolling resampling and artificial injection of anomalies.

The acknowledged limitation of this purely data-driven model lies in its heavy reliance on the quality and quantity of historical sensor data, which can result in lower interpretability compared to physics-based models. Furthermore, the model’s generalizability across various geological and operational conditions requires further rigorous testing. Future research should focus on developing hybrid models that combine the predictive power of deep learning with the physical constraints of the strata–support interaction.

## 5. Conclusions

This study developed a GRU model that, when combined with AM, is used to predict hydraulic support resistance and assess associated formation pressure risks. The methodology encompassed data preprocessing, multi-source feature integration, dynamic weight allocation via the AM, model training, and subsequent risk assessment based on prediction results. The main findings and conclusions are summarized below.

The GRU-AM model demonstrated effective performance in predicting support resistance for both single and multiple supports. For single-support prediction, the model’s predicted curve closely tracked resistance fluctuations, capturing both stable periods and sharp mutation phases. In multi-support prediction scenarios, the average predicted resistance value for various supports was consistently within 0.2% of the actual measured resistance, indicating high prediction fidelity.

The integration of the AM contributed to enhanced prediction accuracy by dynamically weighting multi-source input features. Comparative analysis against established LSTM and CNN models showed that the GRU-AM model achieved favorable results across all key metrics (RMSE, MAE, MAPE, and PCC). Notably, the Pearson correlation coefficients (*R*) for the GRU-AM model at prediction step lengths of 1, 5, and 10 min were 0.9701, 0.9452, and 0.9234, respectively, systematically exceeding the performance of both the LSTM and CNN baseline models.

The GRU-AM model exhibited reliable strata pressure anomaly identification capability across varying prediction horizons. At the 1 min step length, the model achieved high diagnostic performance, yielding F1 scores of 0.9851 (normal risk), 0.9206 (low risk), 0.9908 (medium risk), and 1.0000 (high risk), with an overall Accuracy of 0.9741. While prediction performance metrics naturally decreased with increased step length, the GRU-AM model maintained a performance advantage. Specifically, at the 10 min step length, its Accuracy (0.9234) surpassed that of LSTM (0.9087) and CNN (0.8812), demonstrating greater robustness in long-term sequence prediction.

In practical field validation, the GRU-AM model was successfully applied to mine pressure prediction, showing a prediction Accuracy of 0.9701 at the 1 min step length. The model demonstrated consistent stability and high accuracy across different tested geological conditions, suggesting strong practical utility for real-time monitoring and proactive risk management in hydraulic support systems.

From a practical application perspective, the high-precision abnormal roof weighting identification and prediction method provided by the GRU-AM model moves beyond the limitation of sensors serving merely as data acquisition tools, offering theoretical support and practical instances for the development of the smart sensor field. This framework upgrades the hydraulic support sensing system from simple data logging to a real-time diagnostic and early warning platform. By reliably predicting pressure anomalies (low, medium, and high risk) at different time steps, this study provides the basis for designing new-generation electro-hydraulic control systems, enabling the automatic adjustment of support resistance before critical pressure events occur. This transition from reactive control to predictive control significantly enhances the safety and efficiency of longwall mining operations.

## Figures and Tables

**Figure 1 sensors-25-07336-f001:**
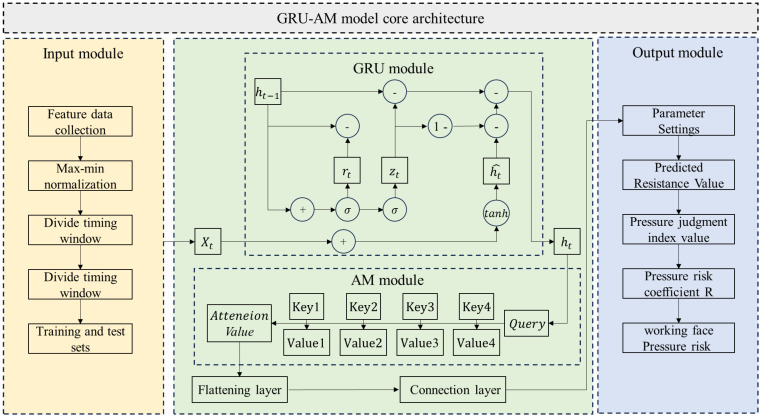
Model architecture of abnormal pressure identification method for hydraulic support working.

**Figure 2 sensors-25-07336-f002:**
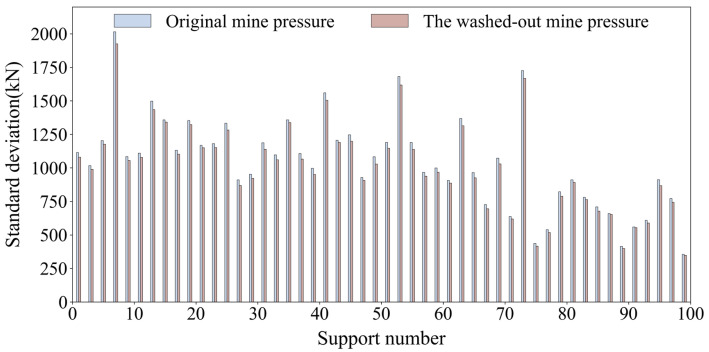
Comparison of standard deviations of each support after mine pressure data cleaning.

**Figure 3 sensors-25-07336-f003:**
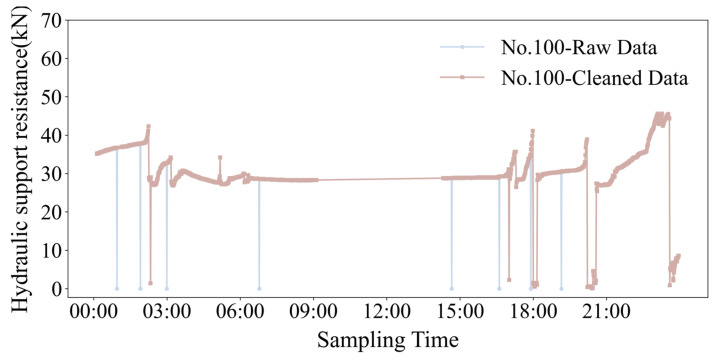
Comparison of the effects after cleaning the resistance value data of 100 brackets.

**Figure 4 sensors-25-07336-f004:**
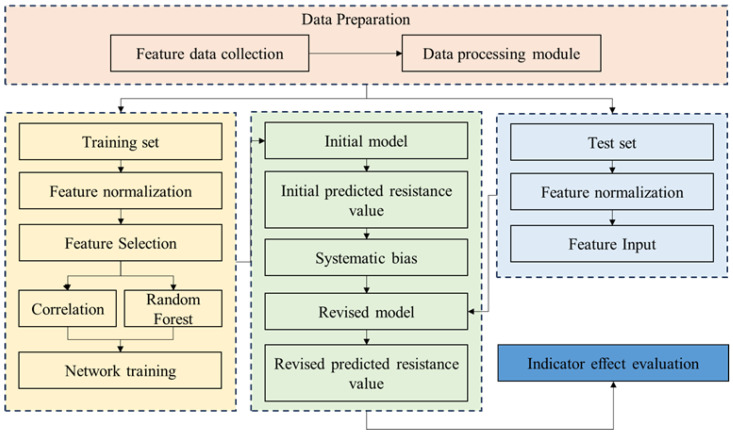
Flowchart of hydraulic support resistance prediction model.

**Figure 5 sensors-25-07336-f005:**
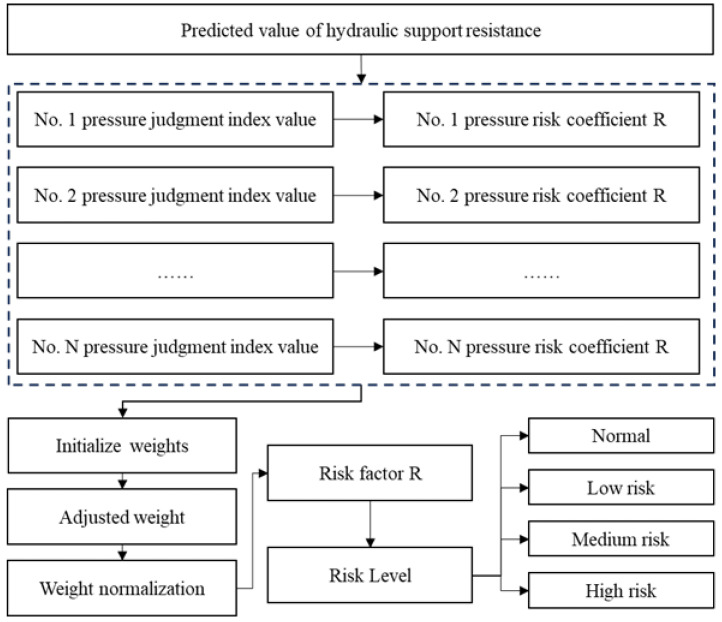
Comprehensive risk assessment model flow chart.

**Figure 6 sensors-25-07336-f006:**
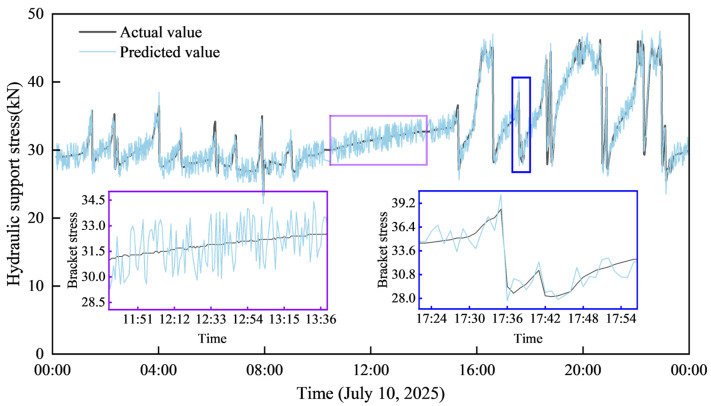
Prediction effect of resistance value of No. 100 bracket.

**Figure 7 sensors-25-07336-f007:**
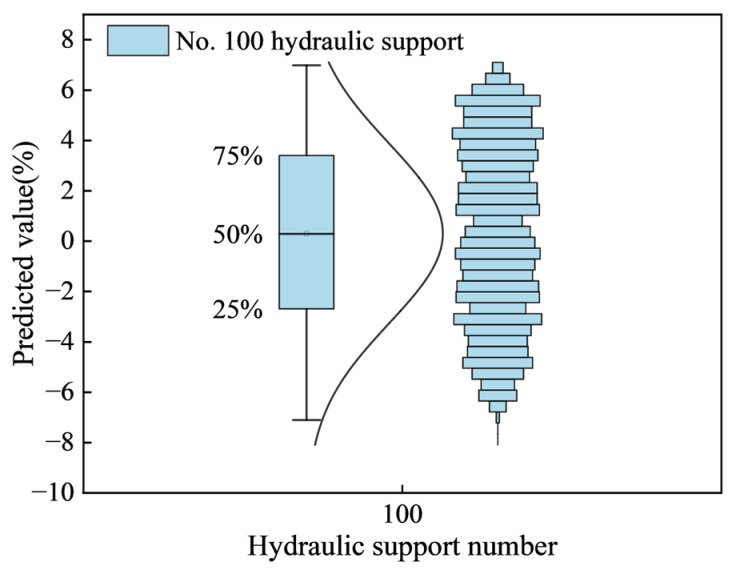
Box plot of relative deviation of the predicted value of No. 100 bracket.

**Figure 8 sensors-25-07336-f008:**
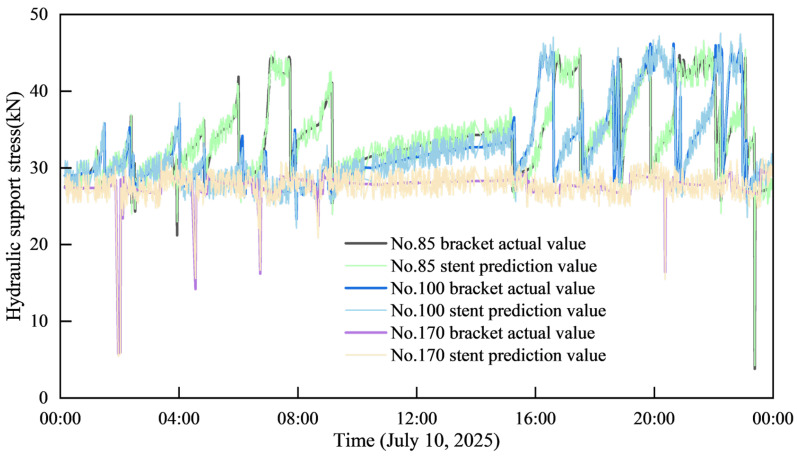
Comparison of the prediction effects of resistance values of multiple supports.

**Figure 9 sensors-25-07336-f009:**
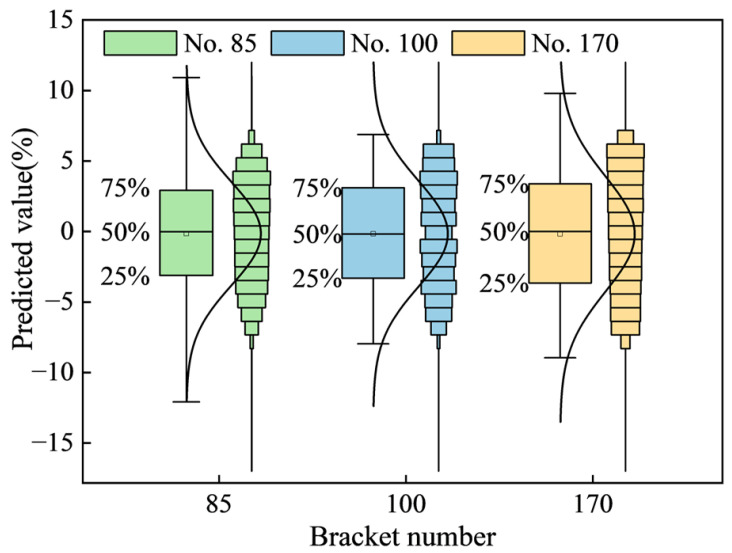
Comparison chart of fluctuation box of predicted values of multiple supports.

**Figure 10 sensors-25-07336-f010:**
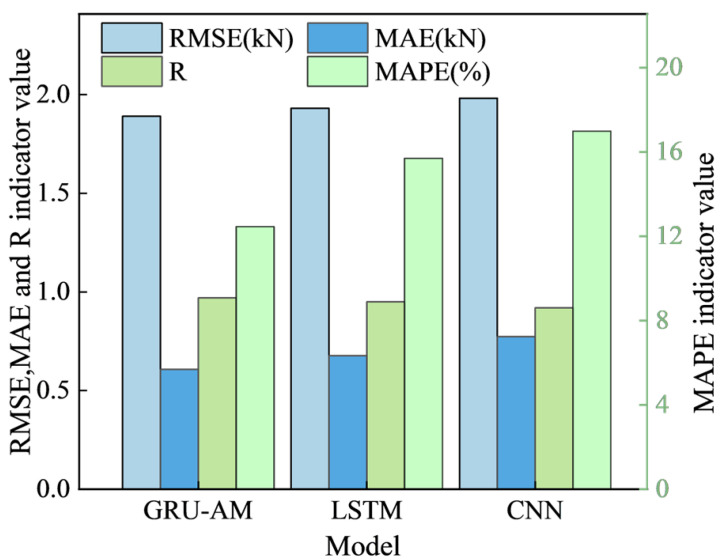
Forecast value fluctuation line chart Prediction results of each model under different time steps.

**Figure 11 sensors-25-07336-f011:**
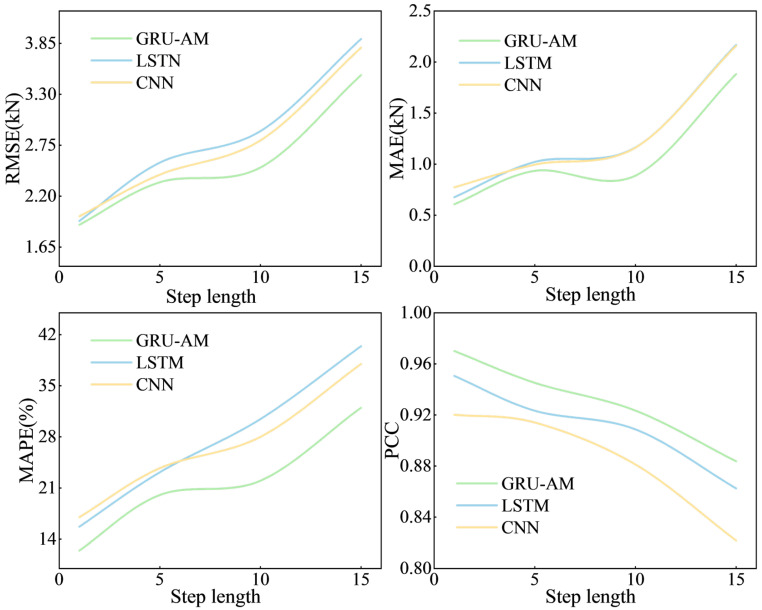
Schematic diagram of prediction effects of different step length models.

**Figure 12 sensors-25-07336-f012:**
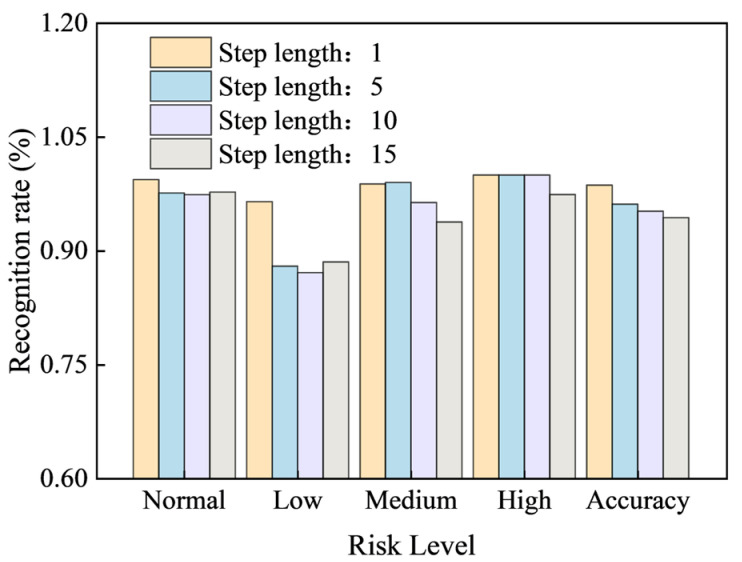
State recognition rate of different step length bracket numbers in 58 cycles.

**Figure 13 sensors-25-07336-f013:**
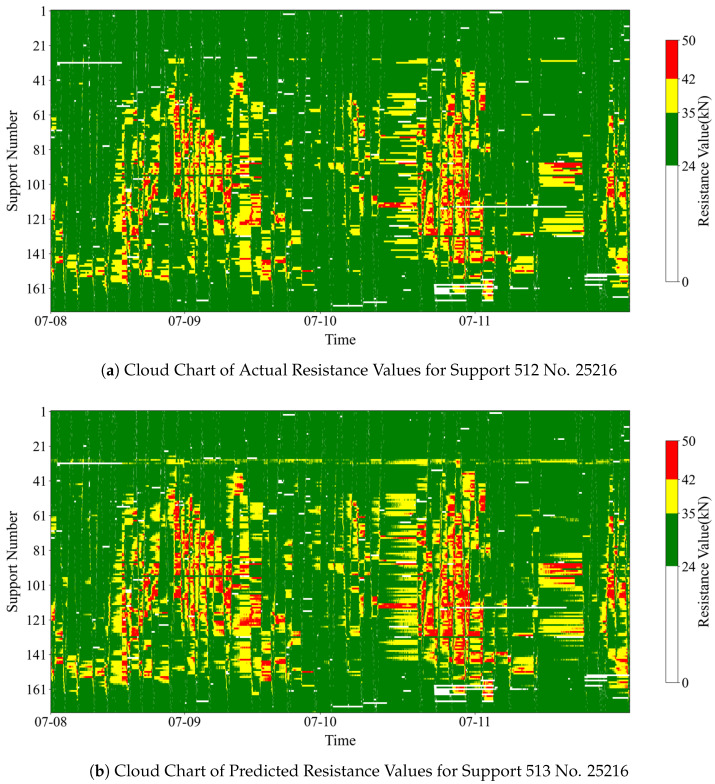
Support resistance value cloud chart.

**Figure 14 sensors-25-07336-f014:**
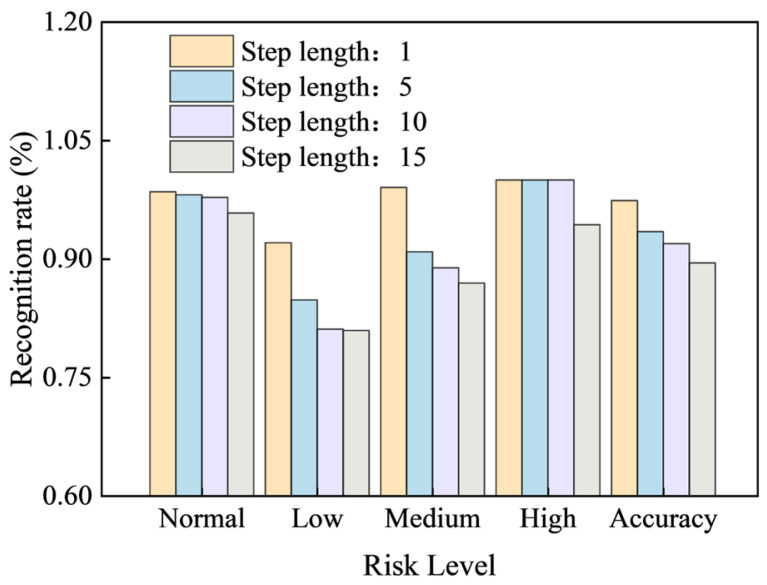
Risk status recognition rate of all stent numbers at different step lengths.

**Figure 15 sensors-25-07336-f015:**
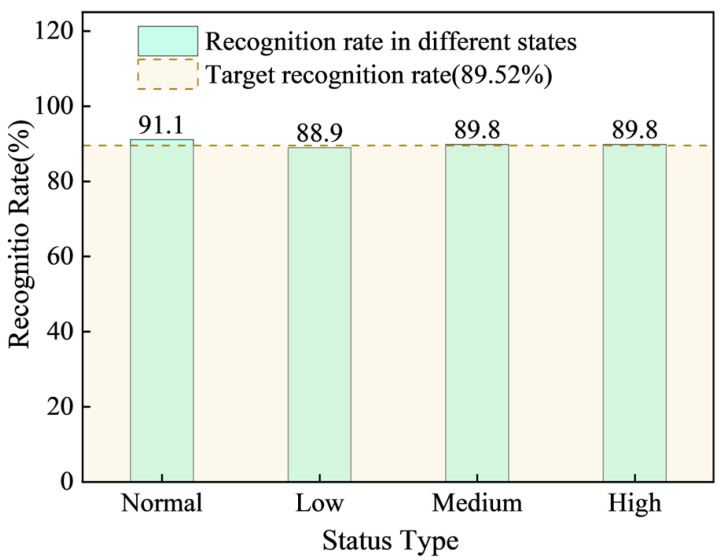
Abnormal pressure identification at a mine working face in Baoji City, Shaanxi Province.

**Table 1 sensors-25-07336-t001:** The advantages and disadvantages of previous stress prediction and identification models.

Model	BP	CNN	LSTM
Time-Series Modeling Capability	No time memory and relies on the current input	Skilled at capturing local temporal correlations	Effective in describing long-term dependencies
Periodic capture capability	Insensitive to the periodicity of the data	Sensitive only to short term cycles	Can handle partial periodicity
Computational Efficiency	Simple structure, fast computation	Good parallelism, relatively fast training	High computational complexity, slow training
Anomaly Detection and Robustness	Highly sensitive to noise and single-feature anomalies, high false positive rate	Sensitive to abrupt changes, weak long-term anomaly recognition	More sensitive to complex anomalies, prone to overfitting

**Table 2 sensors-25-07336-t002:** Comprehensive risk score range of working face pressure.

Comprehensive Risk Score	Risk Level	Response Measures
$S_{\mathrm{risk}}\leq 0.25$	Normal	Routine monitoring
$0.25<S_{\mathrm{risk}}\leq 0.5$	Low Risk	Routine monitoring
$0.5<S_{\mathrm{risk}}\leq 0.75$	Medium Risk	Sound and light alarms and reduction in mining speed to 50%
$S_{\mathrm{risk}}>0.75$	High Risk	Emergency shutdown and reinforcement of support structures

**Table 3 sensors-25-07336-t003:** Working face mine pressure monitoring core indicator data table (2 July 2025).

Bracket ID	Time	Bracket Resistance	Resistance Increase Rate	Time-Weighted Average Resistance Value	Working Face Advancement	Pressing Aver.	Non-Pressing Aver.	Number of Cuts	End-of-Cycle Resistance
1	03:47:52	24.3	0.123	0.635	5.26	43.745	37.25	7	44.242
2	07:44:39	28.1	0.052	0.239	7.58	49.507	36.45	7	50.093
3	10:28:52	27	0.192	0.583	3.79	47.32	27.34	8	47.911
4	15:35:52	28.7	0.297	0.398	21.74	45.987	27.78	8	46.056
5	18:47:52	28.8	0.072	0.257	23.51	41.56	30.89	9	41.597
6	19:37:53	28.3	0.083	0.345	8.23	41.56	37.25	9	41.823
7	22:56:53	28.3	0.016	0.723	6.26	40.581	37.25	10	41.295
8	03:55:54	27.9	0.095	0.685	31.14	48.662	25.87	10	48.688
9	04:34:52	28.2	0.183	0.575	12.8	46.011	35.68	10	46.236
10	06:08:53	28.1	0.115	0.639	6.55	47.081	37.56	11	47.314

**Table 4 sensors-25-07336-t004:** Prediction results of different algorithms under 1 min step size.

Step Length	Model	RMSE (kN)	MAE (kN)	MAPE (%)	PCC
1	GRU-AM	1.8901	0.6074	12.4476	0.9701
	LSTM	1.9314	0.6766	15.6915	0.9506
	CNN	1.9816	0.7734	16.9812	0.9202

**Table 5 sensors-25-07336-t005:** Model prediction effects under different prediction step lengths.

Step Length	Model	RMSE (kN)	MAE (kN)	MAPE (%)	PCC
5	GRU-AM	2.3494	0.9343	20.0326	0.9452
	LSTM	2.5613	1.0217	23.1449	0.9233
	CNN	2.4314	0.9951	23.7136	0.9141
10	GRU-AM	2.5079	0.8877	21.9828	0.9234
	LSTM	2.9016	1.1638	30.4265	0.9087
	CNN	2.8012	1.1615	27.9897	0.8812
15	GRU-AM	3.5065	1.8815	31.9872	0.8837
	LSTM	3.8974	2.1679	40.4213	0.8625
	CNN	3.8029	2.1590	37.9838	0.8217

**Table 6 sensors-25-07336-t006:** Abnormal pressure identification results of stent No. 100 at different times in the future.

Prediction Duration	Classification	Truth Normal	Truth Low Risk	Truth Medium Risk	Truth High Risk
next 1 min	Normal	1136	5	0	0
	Low risk	9	206	0	0
	Medium risk	0	1	42	0
	High risk	0	0	0	14
next 5 min	Normal	1318	15	0	0
	Low risk	50	238	0	0
	Medium risk	0	0	51	0
	High risk	0	0	0	17
next 10 min	Normal	1152	6	1	0
	Low risk	11	209	3	0
	Medium risk	3	1	39	0
	High risk	0	0	0	14
next 15 min	Normal	1471	15	0	0
	Low risk	52	267	6	0
	Medium risk	0	0	50	0
	High risk	0	0	0	18

**Table 7 sensors-25-07336-t007:** Evaluation indicators of No. 100 stent at each step length.

Step Length	Normal F1	Low Risk F1	Medium Risk F1	High Risk F1	Accuracy
1	0.9939	0.9649	0.9882	1	0.9867
5	0.9763	0.8799	0.9903	1	0.9616
10	0.9742	0.8714	0.9639	1	0.9524
15	0.9776	0.8854	0.9381	0.9744	0.9439

**Table 8 sensors-25-07336-t008:** Area ratio of different stress regions.

Regional Area Share	Green (%)	Yellow (%)	Red (%)
Actual value cloud chart	82.47	12.29	3.42
Prediction value cloud chart	80.26	13.66	4.31

**Table 9 sensors-25-07336-t009:** Abnormal pressure identification results of multiple stents at different times in the future.

Prediction Duration	Classification	Truth Normal	Truth Low Risk	Truth Medium Risk	Truth High Risk
next 1 min	Normal	1422	13	0	0
	Low risk	61	261	0	0
	Medium risk	0	0	55	0
	High risk	0	0	0	18
next 5 min	Normal	1366	5	0	0
	Low risk	12	49	1	0
	Medium risk	0	0	5	0
	High risk	0	0	0	2
next 10 min	Normal	1707	7	0	0
	Low risk	1	43	1	0
	Medium risk	0	0	4	0
	High risk	0	0	0	2
next 15 min	Normal	1399	20	2	0
	Low risk	80	250	3	1
	Medium risk	4	4	50	0
	High risk	0	0	0	15

**Table 10 sensors-25-07336-t010:** Evaluation indicators of multiple stents at different step lengths.

Step Length	Normal F1	Low Risk F1	Medium Risk F1	High Risk F1	Accuracy
1	0.9851	0.9206	0.9908	1	0.9741
5	0.9813	0.848	0.9091	1	0.9346
10	0.9781	0.8111	0.8889	1	0.9195
15	0.9582	0.8094	0.8696	0.9434	0.8951

**Table 11 sensors-25-07336-t011:** GRU-AM prediction model performance at different prediction step sizes.

Step Length	RMSE (kN)	MAE (kN)	MAPE (%)	PCC
1	1.8901	0.6074	12.4476	0.9701
5	2.3494	0.9343	20.0326	0.9452
10	2.5079	0.8877	21.9828	0.9234
15	3.5065	1.8815	31.9872	0.8837

**Table 12 sensors-25-07336-t012:** Absolute error distribution of mine pressure prediction in similar geological working faces.

Step Length	MAE (kN)	Sample Size	Proportion
1	MAE ≤0.5	255	12.32%
	0.5<MAE≤1	925	44.69%
	1<MAE≤1.5	480	23.19%
	1.5<MAE≤2	212	10.24%
	MAE>2	198	9.57%

**Table 13 sensors-25-07336-t013:** Abnormal pressure identification at a mine working face in Baoji, Shaanxi Province, China.

Date	Identify Low-Risk Numbers	Identify Medium-Risk Numbers	Identify High-Risk Numbers
2 July 2025	5	0	0
3 July 2025	9	0	0
4 July 2025	4	2	0
5 July 2025	3	2	2
6 July 2025	13	1	1
7 July 2025	7	0	0
8 July 2025	3	0	0
9 July 2025	2	2	0
10 July 2025	13	3	1
11 July 2025	13	1	0

## Data Availability

Data are contained within this article.

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
