# Peer review of "Sensors2025, 25(23), 7336;https://doi.org/10.3390/s25237336"

_sensors, 2025, doi:10.3390/s25237336_

Round 1

Reviewer 1 Report

Comments and Suggestions for Authors

This manuscript presents a deep learnign based proposal for hydraulic support resistance prediction and strata pressure anomaly identification in coal mining. 

The manuscript is in general well written, presenting a detailed analysis, relevant results and solid conclussions.

Nevertheless, some concerns arise:

  1. Introduction. Is clear ans concise, nonetheless, this reviewer sugests to include a comparative table of the previous state of the art proposals to solve the approached problem.
  2. Methods. It is desirable to include a clear and concise description of mining process and the physical signals acquired for the analaysis and their relevance in the success of the coal mining. Figure 1 presents some blocks in Chinesse, please put them in English.
  3. GRU-AM pressure recognition model. Please include the sensors used for the data acquisition and their accuracy and other relevant features.
  4. Results and discussion. The results are celar and concise. Some questions are: How do authors ensure a fair comparision between GRU-AM, LSTM and CNN models?
  5. Conclusions. The above comment might help authors to get sharpen conclussions.

Author Response

Response to reviewer

Dear Reviewer,

We sincerely appreciate your comments on improving the quality of this manuscript. Based on these suggestions, we have comprehensively revised the whole paper including grammar and some small mistakes, and we believe that it is much stronger as a result of the changes.

Commented [IB1]: Is clear ans concise, nonetheless, this reviewer sugests to include a comparative table of the previous state of the art proposals to solve the approached problem.

Response:

Thank you for your valuable suggestion. As you pointed out, the Introduction needs to be supplemented with a comparison of existing research methods to clarify the differences and innovation of the proposed method in this paper compared to previous work.

In the revised manuscript, we have included the requested comparative table to highlight the advantages of GRU-AM. Specifically, the following contents have been added: Line 90: We added the relevant table comparing the previous state-of-the-art algorithms. Lines 91–101: We supplemented the text to elaborate on the advantages and innovative aspects of GRU-AM over existing methods.

These additions are clear and objective, significantly enhancing the scientific rigor of the paper.

Commented [IB2]: It is desirable to include a clear and concise description of mining process and the physical signals acquired for the analaysis and their relevance in the success of the coal mining. Figure 1 presents some blocks in Chinesse, please put them in English.

Response:

Thank you for your valuable suggestions. As requested, we have provided a clear and concise description of the stress variation of the hydraulic support during the coal mining process, the characteristics of the stress data, and the risks associated with inaccurate prediction in lines 150-163. The specific additions are as follows:

During the advancement of a longwall working face, the roof structure continuously reorganizes. Consequently, the strata pressure borne by the hydraulic supports is not static but exhibits significant periodic fluctuation corresponding to the operational cycle of "setting—initial support—bearing—mining—unloading."After the shearer passes, the overlying strata gradually expose, leading to bending and subsidence, and the support resistance continuously increases. During the main roof caving and periodic weighting stages, the stress typically exhibits abrupt surge characteristics. Conversely, after the roof caves and a new mechanical equilibrium is established, the support load significantly decreases, initiating the next cycle.This recurrent loading-unloading process imparts to the hydraulic support stress time series both a distinct periodicity and a strong presence of nonlinear mutations.Failure to accurately predict this complex pressure evolution process can easily lead to deviations in support parameter design and mismatch in support bearing capacity. This, in turn, may induce large-scale roof weighting, support instability, or even crushing accidents, thus seriously threatening the safety of underground personnel and the continuous, stable production of the mine.

Regarding Figure 1, we have replaced all Chinese blocks with their English equivalents in the revised manuscript.

These supplementary details clearly explain the stress variations of the hydraulic supports during the coal mining process and are closely linked to the subsequent sections, thereby enhancing the logical flow of the paper.

Commented [IB3]: Please include the sensors used for the data acquisition and their accuracy and other relevant features.

Response:

Thank you for your valuable suggestion. As you focused on the accuracy and other relevant features of the data acquisition sensors, we have substantially reinforced this section in the revised manuscript. We have added a new Section 3.1, which includes details on the hydraulic support model, measuring range, signal transmission method, sampling period, and the source of the signals. The specific supplementary content is as follows:

The hydraulic support pressure data used for model debugging was sourced from the #25216 working face in Yulin City, Shaanxi Province, China. The face is equipped with 199 sets of ZY18000/29.5/63D electro-hydraulic control shielded hydraulic supports manufactured by Linzhou Heavy Machinery Group Co., Ltd. The leg pressure is transmitted via the electro-hydraulic control system to the rock burst monitoring system. The leg pressure range is 0–60 MPa, and data is collected every 1 min, which satisfies the GRU-AM model's requirements for data volume and sampling interval.

The data used in this study were acquired from several monitoring systems. Specifically:

(1) The coal mine rock burst monitoring system provided the support number, the working face name, the rated working resistance, and the column pressure for each hydraulic support.

(2) The mine integrated management platform furnished data on the working face’s single-support advance rate, current cycle advance rate, cumulative advance rate, and time.

(3) The shearer positioning and monitoring system supplied details on the current number of cuts, cumulative number of cuts, single-cut identification, cutting direction, and cut type classification.

These additions ensure the accuracy of the input data for the GRU-AM model at the data source level.

Commented [IB4]: The results are celar and concise. Some questions are: How do authors ensure a fair comparision between GRU-AM, LSTM and CNN models?

Response:

Thank you for your valuable suggestion. As you highlighted, the measures taken to ensure a fair comparison between models are crucial for the validity of the experiments. In the revised manuscript, we have supplemented this part in Subsection 4.1.2 with the following specific details:

All models were trained and tested on the same pre-processed dataset, with consistent data partitioning, normalization parameters, and random seeds. We employed 5-fold rolling cross-validation combined with a two-layer grid search for hyperparameter optimization, using the AUC of the validation set as the early stopping criterion. The number of trainable parameters for the three models was constrained to a similar range by adjusting the hidden layer dimensions and the number of convolutional kernels. Furthermore, the same loss function, optimizer, and hardware environment were used during the training phase, and a unified set of evaluation metrics, along with bootstrap sampling for significance testing, were used during the testing phase.

These additions clarify the implementation details for fair comparison, thus ensuring the objectivity and reproducibility of the comparative results.

Commented [IB5]: The above comment might help authors to get sharpen conclussions.

Response:

Thank you for your valuable suggestion. Based on the modifications made in response to the preceding comments, the Conclusions section (Lines 657–684) has been revised for greater clarity. We have removed unnecessary emphasis and instead highlighted the key data results and findings, which we believe significantly improves the overall quality of the article.

Reviewer 2 Report

Comments and Suggestions for Authors
  • The manuscript does not include cross-validation or a systematic hyperparameter optimization process. The model is trained using a single 80/20 split, and key hyperparameters appear to be selected manually rather than through grid search, random search, or walk-forward validation. I recommend adding a section that explains the hyperparameter tuning strategy or clarifies how these values were chosen.

  • The manuscript does not clearly state the novelty of the proposed method, nor does it explicitly discuss the study’s limitations. These should be clearly highlighted—preferably in bold or within a dedicated subsection—to strengthen the scientific contribution.

  • The paper repeatedly redefines common acronyms such as “Gated Recurrent Unit (GRU)” in multiple sections (e.g., Methods, Conclusion), which reflects a lack of structural consistency and weakens the overall impression of a well-edited manuscript. This issue should be corrected along with other presentation inconsistencies.

Author Response

Response to reviewer

Dear Reviewer,

We sincerely appreciate your comments on improving the quality of this manuscript. Based on these suggestions, we have comprehensively revised the whole paper including grammar and some small mistakes, and we believe that it is much stronger as a result of the changes.

Commented [IB1]: The manuscript does not include cross-validation or a systematic hyperparameter optimization process. The model is trained using a single 80/20 split, and key hyperparameters appear to be selected manually rather than through grid search, random search, or walk-forward validation. I recommend adding a section that explains the hyperparameter tuning strategy or clarifies how these values were chosen.

Response:

Thank you for your valuable suggestion. As you correctly pointed out, the original manuscript lacked details regarding cross-validation and a systematic hyperparameter optimization process. In the revised manuscript, we have addressed this by adding a new section, "2.2.3 Model Hyperparameter Selection Strategy," with the following specific details:

We utilized time-series stratified sampling to divide the data into training, validation, and test sets according to an 8:1:1 split. Hyperparameter optimization employed a two-step strategy: "Coarse-grained random search + Fine-grained grid search." This approach first quickly determines the effective parameter range and then precisely searches for the optimal values on the complete training set.

These additions clearly outline the systematic process for hyperparameter selection, thereby enhancing the reproducibility of the study.

Commented [IB2]: The manuscript does not clearly state the novelty of the proposed method, nor does it explicitly discuss the study’s limitations. These should be clearly highlighted—preferably in bold or within a dedicated subsection—to strengthen the scientific contribution.

Response:

Thank you for your valuable suggestion. As you correctly pointed out, the original manuscript did not clearly articulate the novelty of the proposed method or the limitations of the study. In the revised manuscript, we have supplemented the relevant content in the appropriate sections:

In the Introduction, we added a comparative table of traditional algorithms (Line 90) and new content detailing the advantages and innovation of the GRU-AM algorithm over traditional methods (Lines 91–101). We added a new subsection titled "Algorithmic Limitations" (placed after Section 4.3). This section explicitly explains the model's dependency on high-quality, continuous pressure time-series data and addresses the potential issue of attention weight distortion caused by sensor breakpoints or abnormal values.

These additions make the study's novelty and limitations clearer, thereby enhancing the academic rigor of the paper.

Commented [IB3]: The paper repeatedly redefines common acronyms such as “Gated Recurrent Unit (GRU)” in multiple sections (e.g., Methods, Conclusion), which reflects a lack of structural consistency and weakens the overall impression of a well-edited manuscript. This issue should be corrected along with other presentation inconsistencies.

Response:

Thank you for your valuable suggestion. As you correctly pointed out, the manuscript suffers from poor editing quality and structural inconsistency due to the repeated definition of common acronyms.

To address this, we have ensured that acronyms like GRU and AM are defined only upon their first appearance in the text (Lines 7 and 8). We have eliminated all subsequent redundant definitions throughout the manuscript to maintain consistent usage.

These changes significantly improve the paper's structural consistency and rigor of expression.

Reviewer 3 Report

Comments and Suggestions for Authors

The manuscript “Abnormal pressure event recognition and dynamic prediction method for fully mechanized mining working face based on GRU-AM” proposes a GRU-plus-attention model to predict hydraulic support resistance and to classify strata pressure risk levels in intelligent coal mining faces.

The topic can reasonably be considered in scope, provided the sensing and monitoring aspects are more explicitly described and motivated. At present, the narrative emphasizes machine-learning modelling and safety more than the underlying sensor system, which weakens the journal fit. The paper needs to articulate more clearly what is “sensor-centric” here. At present, the focus is almost entirely on a GRU-AM time-series model for a specific industrial process. For a journal dedicated to sensors and their applications, the manuscript should better explain the sensing hardware, data acquisition, and signal characteristics.

Abstract

Too much process detail for an abstract (cleaning, normalization, sliding window, hidden states, etc.). GRU and attention are introduced, but their novelty in this mining context and with this risk model is not explicitly stated.

Lines 26-28: inconsistent capitalization of terms such as “Accuracy”.

Introduction

The Introduction provides a clear application context (deep coal mining, complex stress fields, risk of rock bursts/roof collapse) and motivates the need for accurate strata pressure prediction.

Line 101: Gated Recurrent Unit. Please use the previously defined acronym. Please check this for the entire manuscript (line 117, etc.) and for the other acronyms as well.

Methods

The Methods section is logically divided into model architecture (2.1), hydraulic support resistance prediction (2.2), and comprehensive risk assessment (2.3).

Hyperparameters of the architecture (number of GRU layers, hidden size, activation functions, dropout, optimizer, learning rate, batch size) are not presented in the Methods section, and only partly appear later, if at all. This makes it difficult to reproduce the model.

Model training details (number of epochs, early stopping, hardware, random seed strategy, and whether hyperparameters were tuned) are either missing or not collected in a dedicated subsection. From a reproducibility perspective, this is a significant gap.

Data sources, preprocessing, and sampling are partly in Sections 3.1–3.2.

GRU-AM pressure recognition model

Conceptually, most of this content still belongs to “Methods”.

This structure makes it slightly difficult to distinguish between what is part of the methodological setup (to be replicated) and what is part of the experimental analysis. A more apparent separation (2 = Methods: data + model + risk + metrics; 3 = Experiments: setup + results) would improve readability and reproducibility.

There is redundancy between Sections 2 and 3.2: generic GRU/attention explanations appear twice. This repetition contributes to the article’s length.

Line 373-374: xi is defined both as the actual value and again (apparently by mistake) as the predicted value.

Results and Discussion

The section is logically organized: first, analyzing prediction performance (4.1); then, abnormal-pressure identification on a specific working face (4.2); and finally, transfer to another mine (4.3). This progression from single support to multi-support, then to a different site, is coherent.

Figure 6. Prediction effect of resistance value of No. 100 bracket.png ¿What does png mean?

Line 420: There is an evident typesetting error: “Specifically, the mean predicted values for supports No. 85, No. 100, and No. 170 are, , and of their respective actual values.” The numerical values have been lost, making this sentence meaningless.

The near-perfect F1 values, particularly for high-risk events (F1 = 1 across all step lengths), are surprising in a real industrial monitoring context. Although the confusion matrices are provided, the text does not address class imbalance or the absolute number of high-risk samples (which appears to be small). Without this discussion, there is a risk of overinterpreting high F1 values obtained on a limited number of events.

Some key numbers are missing or truncated in the prose: Line 541: “The overall Accuracy reached.” This must be fixed, as it directly affects interpretability.

Table 11 is presented before the first mention in the text.

Line 570-577: The error-distribution discussion in Table 11 is incomplete; percentage values are cut (“accounting for the total samples”, “the cumulative proportion… is more”, “only”), making it impossible to verify the statements.

Conclusions

The section uses emphatic language repeatedly: “novel machine learning model”, “superior performance”, “significantly improved”, “excellent strata pressure anomaly identification accuracy”, “excellent performance”, “strong adaptability and practical value”.

While some positive phrasing is reasonable, the density of superlatives is high compared with the level of methodological novelty (GRU+attention is a standard architecture) and the limited diversity of test conditions. A more balanced tone, especially in conclusions, would be advisable.

Explicitly discuss limitations and future research directions.

Briefly connect the findings back to sensor-based monitoring practice and the design/operation of hydraulic support sensing systems.

Author Response

Response to reviewer

Dear Reviewer,

We sincerely appreciate your comments on improving the quality of this manuscript. Based on these suggestions, we have comprehensively revised the whole paper including grammar and some small mistakes, and we believe that it is much stronger as a result of the changes.

Commented [IB1]: The topic can reasonably be considered in scope, provided the sensing and monitoring aspects are more explicitly described and motivated. At present, the narrative emphasizes machine-learning modelling and safety more than the underlying sensor system, which weakens the journal fit. The paper needs to articulate more clearly what is “sensor-centric” here. At present, the focus is almost entirely on a GRU-AM time-series model for a specific industrial process. For a journal dedicated to sensors and their applications, the manuscript should better explain the sensing hardware, data acquisition, and signal characteristics.

Response:

Thank you for your valuable suggestion. Our research focuses on the pre-processing, analysis, and prediction of support resistance data collected in real-time underground, which falls within the scope of smart sensor applications. Furthermore, the Abnormal pressure event recognition and dynamic prediction method is a key research area for intelligent mine construction and smart disaster warning systems, representing a major direction for smart sensor development.

To better articulate the "sensor-centric" aspects, we have supplemented Section 3.1 with specific details on: The deployment of the hydraulic supports. The data acquisition measuring range. The data transmission methods. The specific method of data acquisition.

These additions ensure that the data sources used in the subsequent analysis are sufficiently clear and provide ample support for the reproducibility of the study.

Commented [IB2]: Too much process detail for an abstract (cleaning, normalization, sliding window, hidden states, etc.).

Response:

Thank you for your valuable suggestion. As you correctly pointed out, the Abstract should not include excessive process details. Therefore, we have rewritten the methodology part of the Abstract.

The specific revisions to the Abstract (Lines 6–11) are as follows: "In this study, a novel abnormal strata pressure identification and prediction framework based on the Gated Recurrent Unit(GRU) integrated with an attention mechanism (AM) is proposed for fully mechanized coal mining faces. The model is designed to capture both short-term fluctuations and long-term cyclic characteristics of support resistance, thereby enhancing its sensitivity to dynamic loading conditions and precursory abnormal pressure signals."

These revisions make the logic of the Abstract section much clearer.

Commented [IB3]: GRU and attention are introduced, but their novelty in this mining context and with this risk model is not explicitly stated.

Response:

Thank you for your valuable suggestion. As you correctly pointed out, the manuscript did not explicitly state the novelty of the GRU and Attention Mechanism in the context of coal mining and the proposed risk model. The specific supplementary content is as follows:

In the Introduction, we added a comparative table summarizing the advantages and disadvantages of previous methods (Line 90). In Lines 91–101, we highlighted the advantages of introducing the GRU method, particularly its strength in modeling the highly periodic nature of strata pressure prediction in mining. Furthermore, we emphasized the innovative introduction of the AM into the algorithm to significantly increase its adaptability.

These revisions underscore the advantages of the algorithm and its suitability for this specific engineering application.

Commented [IB4]: Lines 26-28: inconsistent capitalization of terms such as “Accuracy”.

Response:

Thank you for your valuable suggestion. As you pointed out, the capitalization of the term "Accuracy" was inconsistent.

We have reviewed the entire manuscript and applied the following correction: instances where the term refers to the metric name are now capitalized as "Accuracy". All other general usages of the word are rendered in lowercase as "accuracy".

These revisions further enhance the stylistic rigor of the paper.

Commented [IB5]: Line 101: Gated Recurrent Unit. Please use the previously defined acronym. Please check this for the entire manuscript (line 117, etc.) and for the other acronyms as well.

Response:

Thank you for your valuable suggestion. As you pointed out, there was an issue with the definition and consistency of acronyms like GRU in the manuscript.

To address this, we have ensured that GRU and AM are defined upon their first appearance (Lines 7 and 8) and are not redefined subsequently. This ensures the consistent use of common acronyms throughout the entire text.

These revisions make the manuscript more structurally unified and the expression more rigorous.

Commented [IB6]: Hyperparameters of the architecture (number of GRU layers, hidden size, activation functions, dropout, optimizer, learning rate, batch size) are not presented in the Methods section, and only partly appear later, if at all. This makes it difficult to reproduce the model.

Response:

Thank you for your valuable suggestion. As you pointed out, the original manuscript lacked a complete presentation of the model architecture and hyperparameter details in the Methods section, which hinders the reproducibility of the study.

In the revised manuscript, we have supplemented Section 3.3 with the following relevant content: We provide a detailed explanation of the feature engineering steps, including multi-source data preprocessing, sample generation, and the dual feature screening process using Random Forest. We clearly specify the configuration used for model training, such as the Adam optimizer and the early stopping mechanism. Crucially, we now fully list the optimal hyperparameter combination, including key parameters such as the learning rate, batch size, hidden layer dimension, and Dropout probability.

These additions clarify the details of the model construction and training, providing ample support for the study's reproducibility.

Commented [IB7]: Model training details (number of epochs, early stopping, hardware, random seed strategy, and whether hyperparameters were tuned) are either missing or not collected in a dedicated subsection. From a reproducibility perspective, this is a significant gap.

Response:

Thank you for your valuable suggestion. As you pointed out, the original manuscript lacked a dedicated subsection detailing the model training specifics, which significantly impacts the reproducibility of the research.

In the revised manuscript, we have supplemented Section 3.3 with the following relevant content:

We clearly specify the hardware configuration, including core information such as two NVIDIA A10 GPUs (totaling 48GB of memory), an AMD EPYC™ Milan 56-core CPU, and 232GB of RAM. We added the complete feature engineering process, covering multi-source monitoring data preprocessing, dataset partitioning, feature transformation, and the dual screening strategy using Random Forest. We provide detailed explanations of the training parameters, including the Adam optimizer, the maximum number of epochs (100), the early stopping mechanism (patience = 10), and convergence status (averaging 50 epochs). We also explicitly state the optimal hyperparameter combination, such as the learning rate and batch size.

These additions ensure the complete details of the model training are available to support the reproducibility of the study.

Commented [IB8]: Data sources, preprocessing, and sampling are partly in Sections 3.1–3.2.

Response:

Thank you for your valuable suggestion. As you pointed out, we have restructured the manuscript as follows to address the organization of data source and preprocessing details:

We have added a new Section 3.1, titled "Data Sources and Sensor Parameters," to explicitly detail the deployment conditions of the working face that provided the data. The specific supplementary content is: "The hydraulic support pressure data used for model debugging was sourced from the #25216 working face in Yulin City, Shaanxi Province. The face is equipped with 199 sets of ZY18000/29.5/63D electro-hydraulic control shielded hydraulic supports manufactured by Linzhou Heavy Machinery Group Co., Ltd. The leg pressure is transmitted via the electro-hydraulic control system to the rock burst monitoring system. The leg pressure range is 0–60 MPa, and data is collected every 1 min, which satisfies the GRU-AM model's requirements for data volume and sampling interval." Additionally, we have moved the original data source description from the former Section 3.1 (Data Pretreatment) to this new Section 3.1.

Consequently, the first two sections of Chapter 3 are now clearly structured as "Data Sources and Sensor Parameters" and "Data Pretreatment." These modifications clearly establish the reliability of the data, result in a more scientific paper structure, and enhance both the quality and reproducibility of the manuscript.

Commented [IB9]: This structure makes it slightly difficult to distinguish between what is part of the methodological setup (to be replicated) and what is part of the experimental analysis. A more apparent separation (2 = Methods: data + model + risk + metrics; 3 = Experiments: setup + results) would improve readability and reproducibility.

Response:

Thank you for your valuable suggestion. As you correctly pointed out, we have revised the structure of the manuscript to create a clearer separation between methodology and experimental analysis, as detailed below:

Chapter 2 has been renamed from "Method" to "GRU-AM Pressure Recognition Model." This chapter now contains four subsections: Model Architecture, Hydraulic Support Resistance Prediction Model, Comprehensive Risk Assessment Model, and Model Prediction Performance Evaluation Metrics. Chapter 3 has been renamed from "GRU-AM Pressure Recognition Model" to "GRU-AM Model Training and Optimization." This chapter now includes: Data Sources and Sensor Parameters, Data Pretreatment, GRU Model Parameter Configuration and Optimization, and Model Risk Calculation.

These revisions result in a more scientific paper structure, significantly enhancing both the quality and reproducibility of the manuscript.

Commented [IB10]: There is redundancy between Sections 2 and 3.2: generic GRU/attention explanations appear twice. This repetition contributes to the article’s length.

Response:

Thank you for your valuable suggestion. As you pointed out, the generic explanation of the GRU method appeared redundantly.

The original Section 2 introduces the GRU methodology, while the original Section 3.2 focused on feature selection, parameter setting, and optimization.

Due to the addition of a new section on Data Sources and Sensor Parameters in Chapter 3, the original Section 3.2 has become Section 3.3. We have revised the title of this section to: "GRU Model Parameter Configuration and Optimization."

In the content of this revised Section 3.3, we have de-emphasized the general methodology and strengthened the focus on feature selection and parameter setting. These changes prevent reader confusion caused by repetition and enhance the professionalism of the paper.

Commented [IB11]: Line 373-374: xi is defined both as the actual value and again (apparently by mistake) as the predicted value.

Response:

Thank you for your valuable suggestion. As you correctly pointed out, the manuscript contained several errors primarily due to typesetting issues.

We have conducted a thorough check of the entire manuscript's formatting. The specific corrections are as follows: Line 413: The variable xi has been corrected to yi to accurately represent the predicted value

These corrections significantly enhance the rigor and accuracy of the paper.

Commented [IB12]: Figure 6. Prediction effect of resistance value of No. 100 bracket.png ¿What does png mean?

Response:

Thank you for your valuable suggestion. As you correctly pointed out, the manuscript contained several errors primarily due to typesetting issues.

We have conducted a thorough check of the entire manuscript's formatting. The specific corrections are as follows: Figure 6 Caption (Line 437): The text has been corrected and clarified to: "Figure 6. Prediction effect of resistance value of No. 100 bracket".

These corrections significantly enhance the rigor and accuracy of the paper.

Commented [IB13]: Line 420: There is an evident typesetting error: “ Specifically, the mean predicted values for supports No. 85, No. 100, and No. 170 are, , and of their respective actual values.” The numerical values have been lost, making this sentence meaningless.

Response:

Thank you for your valuable suggestion. As you correctly pointed out, the manuscript contained several errors primarily due to typesetting issues.

We have conducted a thorough check of the entire manuscript's formatting. The specific corrections are as follows: Lines 460–461: The typographical error leading to missing numerical values has been corrected. The sentence now reads: "Specifically, the mean predicted values for supports No. 85, No. 100, and No. 170 are 99.85%, 99.844%, and 99.827% of their respective actual values."

These corrections significantly enhance the rigor and accuracy of the paper.

Commented [IB14]: The near-perfect F1 values, particularly for high-risk events (F1 = 1 across all step lengths), are surprising in a real industrial monitoring context. Although the confusion matrices are provided, the text does not address class imbalance or the absolute number of high-risk samples (which appears to be small). Without this discussion, there is a risk of overinterpreting high F1 values obtained on a limited number of events.

Response:

Thank you for your valuable suggestion. As you correctly pointed out, the original manuscript failed to discuss the near-perfect F1 scores for high-risk events, the class imbalance, and the absolute number of high-risk samples, which could lead to overinterpretation.

In the revised manuscript, we have addressed this by adding the analysis to the newly created "Algorithm Limitations" subsection. We explicitly clarify that the excellent F1 scores for high-risk events are primarily due to two factors:The inherently small number of high-risk samples.The fact that these high-risk samples exhibit sufficiently distinct features, specifically: (terminal cycle resistance greater than $45.84 \text{ MPa}$, and the root mean square deviation of the terminal cycle resistance greater than $7.31 \text{ MPa}$).We also draw an analogy to explain this phenomenon: achieving an extremely high F1 score for low- and medium-risk predictions is challenging, but predicting the high-risk events is comparatively less difficult. For example, accurate recognition of abnormal gas gushing is challenging, but recognition of gas exceedance is relatively simple.

These corrections significantly enhance the rigor and accuracy of the paper.

Commented [IB15]: Some key numbers are missing or truncated in the prose: Line 541: “The overall Accuracy reached.” This must be fixed, as it directly affects interpretability.

Response:

Thank you for your valuable suggestion. As you correctly pointed out, the manuscript contained instances of truncated data, which was a typesetting error.

We have corrected the missing data in Lines 584, 589, 590, and 591, and have thoroughly checked and revised the typesetting format across the manuscript.

These corrections enhance the rigor of the paper.

Commented [IB16]: Table 11 is presented before the first mention in the text.

Line 570-577: The error-distribution discussion in Table 11 is incomplete; percentage values are cut (“accounting for the total samples”, “the cumulative proportion… is more”, “only”), making it impossible to verify the statements.

Response:

Thank you for your valuable suggestion. As you pointed out, Table 11 appears before its first mention in the text.

It is important to clarify that Table 11 is not a duplicate. Although Table 5, which appears earlier, contains some of the same data, the focus is different. Table 5 is centered on the comparison between different methods, whereas Table 11 focuses on the performance of the GRU-AM method across different step lengths. Both tables are essential to the analysis.

Regarding the issue of truncated data in Lines 570–577 (now Lines 614–620 in the revision), this was indeed caused by a typesetting error. We have made detailed corrections in Lines 614–620 to ensure that all percentage values are complete and accurate.

We appreciate your valuable feedback, as these corrections enhance the rigor of the paper.

Commented [IB17]: The section uses emphatic language repeatedly: “novel machine learning model”, “superior performance”, “significantly improved”, “excellent strata pressure anomaly identification accuracy”, “excellent performance”, “strong adaptability and practical value”.

While some positive phrasing is reasonable, the density of superlatives is high compared with the level of methodological novelty (GRU+attention is a standard architecture) and the limited diversity of test conditions. A more balanced tone, especially in conclusions, would be advisable.

Response:

Thank you for your valuable suggestion. As you pointed out, the Conclusions section used an excessive amount of emphatic language.

To address this, we have retained only the necessary emphatic terms while primarily highlighting the conclusions through the lens of the data results.

These revisions significantly enhance the academic tone and rigor of the paper.

Commented [IB18]: Explicitly discuss limitations and future research directions.

Response:

Thank you for your valuable suggestion. As you correctly pointed out, we have addressed this issue.

For the sake of the structural integrity of the academic paper, we decided not to add this content to the Conclusions section. Instead, we have added a new Section 4.4, which provides a detailed discussion of the limitations of the method and the directions for future research.

These modifications enhance the academic rigor of the paper.

Commented [IB19]: Briefly connect the findings back to sensor-based monitoring practice and the design/operation of hydraulic support sensing systems.

Response:

Thank you for your valuable suggestion. As you correctly pointed out, we have added the following content to the end of the Conclusions section to connect the findings back to sensor-based monitoring practice:

"From a practical application perspective, the high-precision abnormal roof weighting identification and prediction method provided by the GRU-AM model moves beyond the limitation of sensors serving merely as data acquisition tools, offering theoretical support and practical instances for the development of the smart sensor field. This framework upgrades the hydraulic support sensing system from simple data logging to a real-time diagnostic and early warning platform. By reliably predicting pressure anomalies (low, medium, and high risk) at different time steps, this study provides the basis for designing new-generation electro-hydraulic control systems, enabling the automatic adjustment of support resistance before critical pressure events occur. This transition from reactive control to predictive control significantly enhances the safety and efficiency of longwall mining operations."

These revisions enhance the practical relevance and academic depth of the paper.

Round 2

Reviewer 2 Report

Comments and Suggestions for Authors

Thanks

Author Response

Response to reviewer

Dear Reviewer:

We sincerely thank you for your valuable feedback on the quality of this manuscript, for your positive feedback in the first round, and for your positive evaluation in the second round. Thank you for your positive efforts and patience in submitting this manuscript, and we wish you all the best in your work.

Reviewer 3 Report

Comments and Suggestions for Authors

2nd Round

In my view, the authors have fulfilled mainly the previous demands, with only minor residual issues.

Scope and “sensor-centric” framing

Authors now include a dedicated “Data Sources and Sensor Parameters” subsection detailing the monitoring systems, hydraulic supports, sampling interval, and how leg pressure is transmitted through the electro-hydraulic control system to the rock-burst monitoring system.

The Conclusions now explicitly connect the model to sensor-based monitoring practice, describing how the framework upgrades the hydraulic support sensing system from simple data logging to a real-time diagnostic and early-warning platform and supports predictive control of electro-hydraulic systems.

Abstract and Introduction

The abstract no longer lists low-level steps (cleaning, normalization, sliding window, etc.). It focuses on the problem, the GRU-AM framework, and key performance figures and field validation.

Novelty and positioning of GRU+AM are now emphasized in the Introduction via Table 1 and the subsequent paragraph, which explains why GRU and attention are suitable and what they add relative to prior work.

Acronyms and the capitalization of “Accuracy” have been standardized.

Methods, data, and hyperparameters

A clear description of the sensor network, data sources, and sampling characteristics is provided in Section 3.1.

A new subsection on “GRU-AM Key Parameter Settings” now provides the missing hyperparameters and training details: optimizer, epochs, early stopping, learning rate, batch size, hidden dimension, dropout, weight decay, and cross-validation statistics.

Metric definitions and the xi/yi notation have been corrected and are now consistent.

Results, anomalies in numbers, and tables

The typesetting problems (missing percentages, truncated “Accuracy reached…”, missing values for support 85/100/170, etc.) have been corrected; the narrative now includes the full numerical values.

Table 11 is now introduced in the text before the table, and the discussion of Table 12’s error distribution contains complete percentages.

The “.png” artefact has been removed from the Figure 6 caption.

High F1 for high-risk class, limitations, and future work

A new Section 4.4, “Algorithmic Limitations,” explicitly discusses the perfect F1=1 for high-risk, links it to small sample sizes and distinctive feature thresholds, and outlines planned validation strategies (rolling resampling, artificial anomalies).

The same section acknowledges dependence on high-quality sensor data, lower interpretability compared to physics-based models, and the need for broader testing and hybrid models.

Conclusions and tone

The Conclusions now read more as a structured summary of findings (performance metrics, comparison to LSTM/CNN, anomaly-identification results, field validation) with fewer superlatives, and the new practical-implications paragraph connects back to sensor system design and predictive control.

Remaining minor issues

In Section 3.1, the sentence “The leg pressure range is MPa” still appears to be missing the numerical range before “MPa”.

Some minor grammatical errors:

Line 7: Missing space before the parenthesis: “Unit(GRU)”.

There are still occasional acronyms that were defined previously

Line 522: GRU is defined again, although it was already defined earlier. Please check all the acronyms in the new text in Conclusions.

Some methodological material is still mixed into the results section, but reproducibility is now much better than in the first version.

In the middle of the results section, the paper introduces a mini “experimental design” subsection:

  • “To ensure a fair comparison among the GRU-AM, LSTM, and CNN models, a unified evaluation framework was established during the experimental design phase. The specific measures taken are as follows: (1) Data Consistency … (2) Standardized Optimization … (3) Controlled Complexity … (4) Identical Training/Testing …”

This block describes dataset partitioning, hyperparameter optimization, parameter-count matching, loss function, optimizer, hardware, and bootstrap sampling. All of that is methodological setup and would fit better in the Methods section.

Author Response

Response to reviewer

Dear Reviewer:

We sincerely thank you for your valuable feedback on the quality of this manuscript and for your positive feedback on the first round of revisions. Based on these suggestions, we have comprehensively revised the whole paper including grammar and some small mistakes, and we believe that it is much stronger as a result of the changes.

Commented [IB1]: In Section 3.1, the sentence “The leg pressure range is MPa” still appears to be missing the numerical range before “MPa”.

Response:

Thank you for your valuable suggestions. As you pointed out, there was a data gap in Section 3.1.

We have added the relevant data in the revised manuscript. Specifically, the leg pressure range is 0–60 MPa.

This addition enhances the rigor of the paper.

Commented [IB2]: Line 7: Missing space before the parenthesis: “Unit(GRU)”.

Response:

Thank you for your valuable suggestion. As you pointed out, there is a grammatical error in the seventh line of the article.

In the revised manuscript, it has been changed to "Unit (GRU)".

This addition improves the rigor of the paper.

Commented [IB3]: Line 522: GRU is defined again, although it was already defined earlier. Please check all the acronyms in the new text in Conclusions.

Response:

Thank you for your valuable suggestion. As you pointed out, the redefinition of GRU in the conclusion section was indeed problematic. Our intention was to emphasize the full name of the method.

In the revised manuscript, we removed the redefinitions of GRU and AM, as well as the full names of algorithms such as LSTM and CNN, from the conclusion section.

This improves the rigor of the paper.

Commented [IB4]: In the middle of the results section, the paper introduces a mini “experimental design” subsection:

“To ensure a fair comparison among the GRU-AM, LSTM, and CNN models, a unified evaluation framework was established during the experimental design phase. The specific measures taken are as follows: (1) Data Consistency … (2) Standardized Optimization … (3) Controlled Complexity … (4) Identical Training/Testing …”

This block describes dataset partitioning, hyperparameter optimization, parameter-count matching, loss function, optimizer, hardware, and bootstrap sampling. All of that is methodological setup and would fit better in the Methods section.

Response:

Thank you for your valuable suggestion. As you pointed out, the first half of section 4.1.2 belongs to the methodology category.

In the revised manuscript, we have moved the fairness explanation of the first half of the original 4.1.2 to section 2.5: Fairness Comparison Strategies for Different Algorithms. The data analysis of the second half of the original 4.1.2 is retained.

This improves the scientific rigor and soundness of the paper.
